# Bootstrapping MLLM for Weakly-Supervised Class-Agnostic Object Counting

**Xiaowen Zhang[1], Zijie Yue[1], Yong Luo[2], Cairong Zhao[3], Qijun Chen[1], Miaojing Shi[1,4]***

[1]College of Electronic and Information Engineering, Tongji University
[2]College of Computer Science, Wuhan University
[3]College of Computer Science, Tongji University
[4]State Key Laboratory of Autonomous Intelligent Unmanned Systems

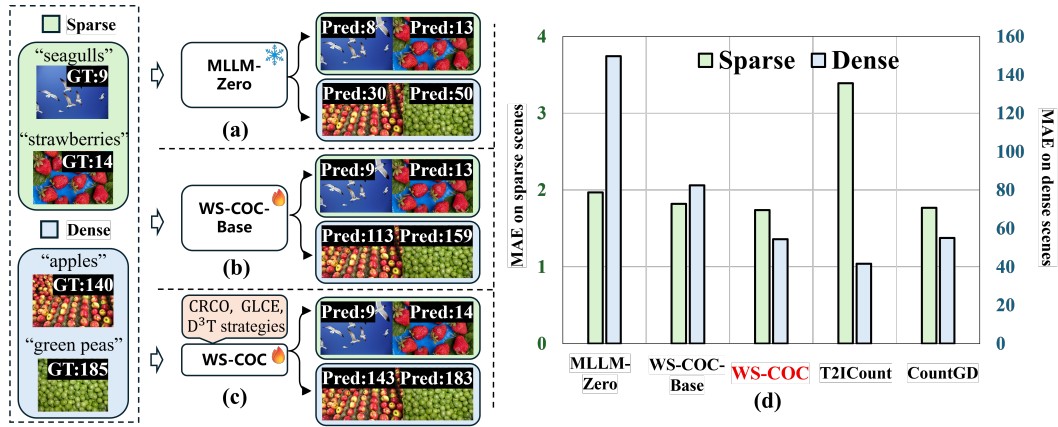

Figure 1: (a)-(c) Visualizations results for MLLM-Zero, WS-COC-Base and WS-COC. (d) MAE results of MLLM-Zero, WS-COC-Base, WS-COC, and two state-of-art fully-supervised counting methods (T2ICount (Qian et al., 2025) and CountGD (Amini-Naieni et al., 2024)) on sparse (up to 20 instances per image) and dense (more than 100 instances per image) object scenes. WS-COC significantly outperforms MLLM-Zero and WS-COC-Base, and yields competitive performance to T2ICount and CountGD. Experiment is on FSC-147 (Ranjan et al., 2021) and MLLM is LLaVA-OneVersion-7B (Li et al., 2024).

## Abstract

Object counting is a fundamental task in computer vision, with broad applicability in many real-world scenarios. Fully-supervised counting methods require costly point-level annotations per object. Few weakly-supervised methods leverage only image-level object counts as supervision and achieve fairly promising results. They are, however, often limited to counting a single category, *e.g.* person. In this paper, we propose WS-COC, the first MLLM-driven weakly-supervised framework for class-agnostic object counting. Instead of directly fine-tuning MLLMs to predict object counts, which can be challenging due to the modality gap, we incorporate three simple yet effective strategies to bootstrap the counting paradigm in both training and testing: First, a divide-and-discern dialogue tuning strategy is proposed to guide the MLLM to determine whether the object count falls within a specific range and progressively break down the range through multi-round dialogue. Second, a compare-and-rank count optimization strategy is introduced to train the MLLM to optimize the relative ranking of multiple images according to their object counts. Third, a global-and-local counting enhancement strategy aggregates and fuses local and global count predictions to improve counting performance in dense scenes. Extensive experiments on FSC-147, CARPK, PUCPR+, and ShanghaiTech show that WS-COC matches or even surpasses many state-of-art fully-supervised methods while significantly reducing annotation costs. Code is available at https://github.com/viscom-tongji/WS-COC.

*Corresponding author: mshi@tongji.edu.cn

## 1 INTRODUCTION

Object counting refers to the task of estimating the number of target instances within a given image. Conventional methods (Radford et al., 2021; Liu et al., 2022a; Amini-Naieni et al., 2023) mainly estimate a density map of an image where the integral over the density map gives the total number of objects. The density map is obtained by convolving Gaussian kernels with point annotations per object in the image. Despite that these fully-supervised methods have led to strong performance across various counting benchmarks (Ranjan et al., 2021; Hsieh et al., 2017; Mundhenk et al., 2016), acquiring point-level annotations is labor-intensive and time-consuming, especially in dense scenes where hundreds or thousands of objects co-occur and even occlude each other.

To alleviate the high annotation cost, few studies (Wang et al., 2024; Yang et al., 2020; Lei et al., 2021) have explored the weakly-supervised learning paradigm. They leverage image-level object counts as the ground truth to learn a mapping from visual features of an image to the total number of target instances in the image. Existing weakly-supervised counting methods are still in early stage and often limited to a single object category, *i.e.* person (Yang et al., 2020; Lei et al., 2021; Xiong et al., 2022). Benefiting from large-scale pretraining on vision-language pairs, recent advances in Multimodal Large Language Models (MLLMs) present a new opportunity for class-agnostic object counting in a text-promptable way, where text prompts specify target categories of interest and are then converted to proper instructions. Combined with corresponding images, these multimodal instructions are fed into the MLLM to auto-regressively predict object counts. This presents a basic object counting pipeline based on the MLLM. We first evaluate this pipeline (*i.e.* MLLM-Zero) without any fine-tuning on a standard object counting benchmark, FSC-147 (Ranjan et al., 2021). As shown in Fig. 1(a): we observe that it indeed produces reasonable estimates in sparse scenarios yet its performance degrades significantly in dense scenes. This can be intuitively explained by the fact that MLLMs are mostly seen with sparse rather than dense object distributions in their pretraining corpora. Given the underlying counting ability of MLLMs, this study aims to explore how to extend such potentials to accurate class-agnostic object counting with minimal fine-tuning cost.

In this paper, we introduce a **W**eakly-**S**upervised **C**lass-agnostic **O**bject **C**ounting network, **WS-COC**, which bootstraps MLLMs for object counting using only image-level count supervision. A naive baseline (*i.e.* WS-COC-Base) is to directly fine-tune the MLLM to predict the object count from the given image. Nevertheless, as shown in Fig. 1(b), learning a direct mapping is difficult due to the absence of object distribution supervision; especially in dense scenes, this still leads to clear underestimation. To address it, we introduce three simple yet effective strategies:

- First, instead of regressing the exact count in one shot, we introduce a *divide-and-discern dialogue tuning* ($D^3T$) strategy to guide the MLLM to judge whether the count falls within a certain range and progressively break down the range from big to small. This multi-step reasoning helps MLLM learn to count from easy to hard. When the range is small enough, we ask the model to predict the actual object count;
- Next, directly optimizing the predicted count against the ground truth can still be challenging due to the modality gap between vision and text. Instead, guiding MLLM to judge the relative count difference between images can be more visually probed. Motivated by this, we introduce a *compare-and-rank count optimization* (CRCO) strategy that trains the MLLM to predict relative ranking of multiple images according to their object counts.
- Last, during inference, to further mitigate the counting bias in dense crowds, we introduce a *global-and-local counting enhancement* (GLCE) strategy. We first predict a global count for the given image, and then partition the input image into smaller sub-images, querying each sub-image independently to obtain local counts. The global and local counts are fused to obtain the final count.

As illustrated in Fig.1(d), through the implementation of these strategies, we significantly enhance the quantity awareness of MLLM, leading to substantial reductions in counting errors, especially under dense conditions. We conduct extensive experiments on four widely-used benchmarks: FSC-147 (Ranjan et al., 2021), CARPK (Hsieh et al., 2017), PUCPR+ (Hsieh et al., 2017), and ShanghaiTech (Zhang et al., 2016). Our method, as the first MLLM-driven weakly-supervised object counting method, yields surprisingly good results on par with state-of-the-art fully-supervised object counting methods (Pelhan et al., 2024; Zhu et al., 2024; Hui et al., 2024).

## 2 RELATED WORK

### 2.1 OBJECT COUNTING

Existing counting methods can be broadly divided into class-specific and -agnostic ones depending on whether they could count arbitrary object categories.

Class-specific object counting aims to predict the number of objects for a specific category, such as persons (Ranasinghe et al., 2024; Guo et al., 2024; Peng & Chan, 2024; Wu & Yang, 2023; Du et al., 2023; Liu et al., 2022b; Shi et al., 2019), vehicles (Hsieh et al., 2017; Mundhenk et al., 2016), cells (Xie et al., 2018), fishes (Sun et al., 2023), *etc.* . Conventional methods are typically detection-based (Liu et al., 2023; Liang et al., 2022; Liu et al., 2019) or regression-based (Ranasinghe et al., 2024; Guo et al., 2024; Peng & Chan, 2024; Wu & Yang, 2023; Li et al., 2023; Wang et al., 2020). The former detects every individual instance with a bounding box and obtains object count as the number of bounding boxes; the latter estimates a density map for a given image and obtains object count by integrating pixel values over the density map. Overall, class-specific methods are inherently restricted to certain object category, limiting their applications.

Recently, class-agnostic counting (Chattopadhyay et al., 2017; Lu et al., 2019; Xu et al., 2023) methods have been proposed to count objects of arbitrary categories within images. A common practice is to provide visual exemplars to indicate the target category (Lu et al., 2019; Liu et al., 2022a; Pelhan et al., 2024), as in CounTR (Liu et al., 2022a), which fuses image and exemplar features for density map estimation. However, since manually selecting exemplars requires extensive labeling cost, an emerging way is to replace them with text prompts that use class names to specify target object categories (Xu et al., 2023; Amini-Naieni et al., 2024; Jiang et al., 2023; Kang et al., 2024; Zhu et al., 2024; Hui et al., 2024; Dai et al., 2024; Qian et al., 2025; Zhai et al., 2025; Shi et al., 2025). For example, VLPG (Zhai et al., 2025) leverages CLIP (Radford et al., 2021) to extract visual embeddings from the given image and textual embeddings from the text prompt, then fuses them via cross-attention for density map estimation.

So far, the majority of object counting methods adopts the fully-supervised learning paradigm, where models are trained with point-level annotations. To reduce the annotation cost, few methods (Yang et al., 2020; Meng et al., 2021; Lei et al., 2021; Xiong et al., 2022; Wang et al., 2024) have explored the weakly-supervised learning paradigm, where only image-level object counts are provided as the ground truth. For example, Xiong et al. (2022) leverage the vision transformer (ViT) to extract multi-scale visual features from images and directly map them to person counts. GCNet (Wang et al., 2024) utilizes a pretrained ResNet (He et al., 2016) to count the most frequently occurring object category in images. These initial explorations, based on CNN or ViT, are limited to class-specific object counting; while we novelly leverage MLLM to achieve class-agnostic object counting.

### 2.2 MULTIMODAL LARGE MODELS

Early multimodal large models, commonly referred to as vision-language models (VLMs), such as CLIP (Radford et al., 2021) and BLIP (Li et al., 2022), learn cross-modal representations through large-scale pretraining on vision-language pairs. They typically perform vision-language contrastive learning and vision-language matching to align vision and text modalities, and have been widely used in multimodal downstream tasks (Jiang et al., 2023; Kang et al., 2024; Monsefi et al., 2024). In recent years, multimodal large language models (MLLMs) (Liu et al., 2024a; Li et al., 2024; Bai et al., 2025) have emerged as an upgrade of VLMs with multimodal reasoning and generative capability. By incorporating large language models (LLMs) as the core semantic understanding and reasoning engine, MLLMs have achieved remarkable success across a broad range of tasks, such as visual question answering (Goyal et al., 2017) and image captioning (Agrawal et al., 2019).

Several studies (Jiang et al., 2023; Qharabagh et al., 2024) have leveraged multimodal large models to address object counting in a text-promptable manner. For instance, CLIP-Count (Jiang et al., 2023), VLPG (Zhai et al., 2025) and VLCounter (Kang et al., 2024) finetune the pretrained VLM, *i.e.* CLIP Radford et al. (2021), for class-agnostic counting. CLIP and other VLMs are mainly discriminative models where object count has to be obtained by devising an additional counting head. Our WS-COC, on the other hand, relies solely on MLLM where object count can be auto-regressively generated. Notably, some VLM-based counting method, such as CrowdCLIP (Liang et al., 2023), adopts a ranking strategy that appears similar to our proposed compare-and-rank count

optimization strategy, but there are key differences. First, Liang et al. (2023) constructs image sets by cropping a single image into sub-images of different sizes, assuming that larger crops contain more objects. In contrast, WS-COC samples different images with varying object counts within the same category. Second, Liang et al. (2023) relies on the contrastive optimization while WS-COC directly generates ranks using the language modeling loss.

# 3 PRELIMINARY

In this section, we first provide a brief background for MLLMs and then devise a baseline model upon them for weakly-supervised object counting.

**Multimodal Large Language Models.** MLLMs, specifically for vision and language, consist of a visual encoder, a projection module, and a LLM. They take the image $I$ and text instruction $T^{\text{inst}}$ as input and output a text response $T^{\text{res}}$. Specifically, the visual encoder extracts visual features from $I$. The projection module transforms these features into ones that can be fed into the LLM. The LLM then performs semantic understanding and reasoning based on the projected visual features and instruction tokens of $T^{\text{inst}}$ to auto-regressively generate $T^{\text{res}}$. MLLMs are trained using a language modeling loss that minimizes the cross-entropy between $T^{\text{res}}$ and ground truth response $T^{\text{gt}}$.

**Baseline.** We devise a baseline model (*i.e.* WS-COC-Base) in which we finetune a MLLM, LLAVA-OneVersion (Li et al., 2024) by default, for weakly-supervised object counting. Specifically, for each image $I$ and its global object count $c$, we construct the text instruction $T^{\text{inst}}$ and the ground truth response $T^{\text{gt}}$ using fixed templates. $T^{\text{inst}}$ is formulated as "*How many [obj] are there in the image?*", while $T^{\text{gt}}$ follows "*a photo of [num] [obj]*", where the [num] token is replaced with $c$ and the [obj] token denotes a specific object category. We then follow the LoRA-based finetuning paradigm (Hu et al., 2022) to optimize the MLLM. WS-COC-Base's performance is shown in Fig.1(d).

# 4 METHOD

Fig.2 illustrates the framework of our proposed WS-COC, in which we bootstrap the MLLM for weakly-supervised object counting using only image-level count supervision. Building upon the baseline model, we introduce three simple yet effective strategies: divide-and-discern dialogue tuning, compare-and-rank count optimization, and global-local counting enhancement.

## 4.1 DIVIDE-AND-DISCERN DIALOGUE TUNING

Learning to predict an absolute object count from an image is difficult for MLLMs, especially without explicit supervision on individual objects. To cope with it, we reformulate the count prediction task as a series of range judgment tasks: instead of regressing the exact count in one shot, we guide the model to iteratively determine whether the count falls within specific ranges through multi-round dialogue. By progressively breaking down the range from coarse to fine, the model is able to learn to count in a curriculum manner, moving from easy to hard.

Specifically, given an image $I$ and its global object count $c$, we set the initial range $[L_1, U_1]$ for the first round dialogue, *e.g.* minimal and maximum object counts $[1, 2000]$ in the FSC-147 dataset (Ranjan et al., 2021). We calculate the midpoint of this range $\tau_1 = \lfloor \frac{(L_1 + U_1)}{2} \rfloor$ and construct the text instruction $Q_1$ using the template "*Are there more than $\tau_1$ [obj] in the image?*", where [obj] token is replaced by the target object category. The ground truth response $R_1^{\text{g}}$ is set to "*yes*" if $c > \tau_1$ and "*no*" otherwise. We then query the MLLM using $I$ and $Q_1$ to predict the response $R_1$, which is optimized against $R_1^{\text{g}}$ using the language modeling loss.

We progressively half the range and repeat the above process for multiple rounds. For example, in the $t$-th round, we update $[L_t, U_t]$ according to the ground truth response $R_{t-1}^{\text{g}}$ of $(t-1)$-th round:

$$[L_t, U_t] = \begin{cases} [\tau_{t-1} + 1, \ U_{t-1}], & \text{if } R_{t-1}^{\text{g}} = \texttt{Yes}, \\ [L_{t-1}, \ \tau_{t-1}], & \text{if } R_{t-1}^{\text{g}} = \texttt{No}. \end{cases} \tag{1}$$

The dialogue terminates when $U_t - L_t$ is lower than $\delta = 0.2 \times c$. Then, the MLLM is asked to predict the actual object count.

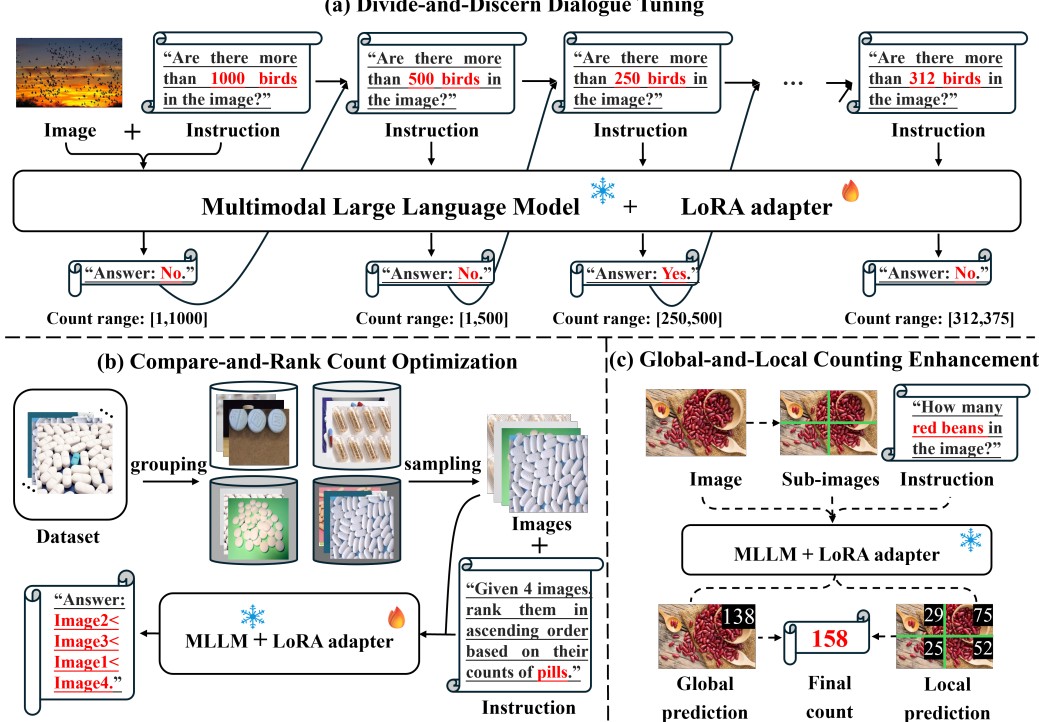

Figure 2: Overview of the proposed WS-COC for weakly-supervised class-agnostic object counting. (a) Divide-and-discern dialogue tuning (Sec. 4.1) guides the MLLM to determine whether the object count falls within a specific range and progressively break down the range through multi-round dialogue. (b) Compare-and-rank count optimization (Sec. 4.2) trains the MLLM to rank images by their object counts. (c) Global-and-local counting enhancement (Sec. 4.3) aggregates and fuses the global and local count predictions to improve the performance in dense scenes during inference.

## 4.2 COMPARE-AND-RANK COUNT OPTIMIZATION

Visual features are high-dimensional representations, whereas the ground truth count is a discrete scalar expressed as a text token. Directly optimizing the predicted count against the ground truth can be challenging due to the modality gap. This makes it difficult for the model to establish a robust mapping. In contrast, guiding the model to judge the relative count difference between images can be intuitively more accessible. Therefore, we propose a compare-and-rank count optimization strategy that trains the model to compare multiple images and optimize their relative ranking according to object counts.

First, we need to sample multiple images for count comparison. Since counting datasets typically exhibit a long-tailed distribution of object counts, dense scenes with large object counts are rare and unlikely to be sampled through random sampling. Therefore, we introduce a simple scheme to cluster images according to their counts. Specifically, we obtain the minimum and maximum counts (i.e. , $[\underline{c}, \overline{c}]$) for each object category in the training set, and then partition $[\underline{c}, \overline{c}]$ into $K$ (4 by default) equal-length intervals. Accordingly, the images of each category are divided into four groups based on intervals in which their object counts fall. During each training iteration, we randomly sample one image from each group to construct an image set $\mathcal{I} = \{I_1, I_2, I_3, I_4\}$, where the corresponding counts inherently satisfy $c_1 < c_2 < c_3 < c_4$. In this way we ensure that both sparse and dense scenes can be covered. We randomly shuffle the order of images in $\mathcal{I}$ to $\tilde{\mathcal{I}}$ and treat it as the visual input to the MLLM.

Next, we construct the associated text instruction $T$ using the template "*Given four images, rank them in ascending order based on their counts of [obj]*", and the ground truth response $T^g$ specifies the correct ranking in the form "*Image $i < \cdots <$ Image $j$*", where "*Image $i$*" and "*Image $j$*" denote

the $i$-th and $j$-th image in $\tilde{\mathcal{I}}$, respectively. We use $T$ and $\tilde{\mathcal{I}}$ to query the MLLM to obtain the response $T^{\mathrm{r}}$, then optimize it with respect to $T^{\mathrm{g}}$ using the language modeling loss.

### 4.3 GLOBAL-AND-LOCAL COUNTING ENHANCEMENT

Once the model is fine-tuned, we observe that it still tends to underestimate object counts in dense scenes, although its performance is much improved compared to that before fine-tuning. Therefore, during the inference stage, to further mitigate the counting bias in dense crowds, we propose to partition the images with dense scenes into smaller sub-images and query the MLLM on each sub-image independently. The local predictions are aggregated and combined with the global prediction to produce the final output.

Specifically, we first query the MLLM to predict the global count $c^{\mathrm{g}}$ for the given image $I$. If $c^{\mathrm{g}}$ is lower than the threshold $c^{\mathrm{h}}$, we directly treat $c^{\mathrm{g}}$ as the final prediction. Otherwise, the image $I$ is considered as a dense scene. In this case, we evenly partition $I$ into $L \times L$ ($2 \times 2$ by default) non-overlapping sub-images and obtain local predictions $\{c_k\}_{k=1}^{L^2}$ from the MLLM by respectively querying each sub-image. We then sum the local predictions to obtain $c^{\mathrm{l}}$. Due to the edge effect when aggregating sub-images, $c^{\mathrm{l}}$ often overestimates the total count compared to $c^{\mathrm{g}}$. Empirically, we obtain the final prediction by simply averaging $c^{\mathrm{l}}$ and $c^{\mathrm{g}}$. In this way, they complement each other in estimating the final count in dense scene (see Sec. 5.3: GLCE).

### 4.4 TRAINING AND INFERENCE

On top of the baseline (Sec. 3), divide-and-discern dialogue tuning and compare-and-rank count optimization are additionally applied only during training, while the global-and-local counting enhancement is activated only during inference. Text instruction for inference is formulated as "*How many [obj] are there in the image?*", where the [obj] token denotes the target object category.

## 5 EXPERIMENTS

### 5.1 EXPERIMENT DETAILS

**Datasets.** Our experiments are primarily conducted on the FSC-147 dataset (Ranjan et al., 2021), which contains 6,135 images spanning 147 object categories. We report the model performance on both the validation and test sets of FSC-147. Following prior works (Jiang et al., 2023; Kang et al., 2024), we also investigate cross-dataset generalization performance by testing the trained model from FSC-147 onto CARPK (Hsieh et al., 2017), PUCPR+ (Hsieh et al., 2017), and ShanghaiTech (Zhang et al., 2016). CARPK consists of nearly 90,000 annotated cars captured from four parking lots via drones, while PUCPR+ contains 16,456 car instances in bad weather. The ShanghaiTech dataset is divided into SHA (482 internet images) and SHB (716 street-view images from Shanghai), covering diverse crowd densities and scene layouts.

**Implementation details.** We adopt LLaVA-OneVision-7B (Li et al., 2024) as the default MLLM and follow MM-RAIT (Liu et al., 2025) to optimize it using the LoRA-based finetuning paradigm: the rank of LoRA adapter is set to 128 with a scaling factor of 256. The count threshold $c^{\mathrm{h}}$ is set to 100. We set $L = 2$ in the global-and-local counting enhancement strategy. The batch size is 4, and training is conducted on a NVIDIA L20 GPU.

**Evaluation metrics.** Following common practice in (Jiang et al., 2023; Kang et al., 2024), we evaluate performance using Mean Absolute Error (MAE) and Root Mean Square Error (RMSE).

### 5.2 COMPARISON WITH STATE-OF-THE-ART METHODS

As shown in Tab.1, we compare the proposed method with eight fully-supervised text-promptable object counting methods and three weakly-supervised counting methods on the FSC-147 dataset. Methods marked with "*" denote weakly-supervised variants that we adapted from their fully-supervised counterparts: we replace the last layers of their decoders, which originally predict density maps, with a linear layer for direct count prediction. Moreover, we denote MLLM-Zero as a variant that directly evaluates the MLLM in the zero-shot setting. Notably, our proposed WS-COC achieves

Table 1: Quantitative comparisons on FSC-147. The best fully-supervised approach is marked in shadow , while the best weakly-supervised approach is marked in **bold**. Methods marked with "*" denote weakly-supervised variants that we adapted from their fully-supervised counterparts.

| Method | Supervision | VAL SET | | TEST SET | |
|---|---|---|---|---|---|
| | | MAE↓ | RMSE↓ | MAE↓ | RMSE↓ |
| ZSOC (Xu et al., 2023) | point-level | 26.93 | 88.63 | 22.09 | 115.17 |
| CLIP-Count (Jiang et al., 2023) | point-level | 18.79 | 61.18 | 17.78 | 106.62 |
| VLCounter (Kang et al., 2024) | point-level | 18.06 | 65.13 | 17.05 | 106.16 |
| CountDiff (Hui et al., 2024) | point-level | 15.50 | 54.33 | 14.83 | 103.15 |
| GroundingREC (Dai et al., 2024) | point-level | 10.06 | 58.62 | 10.12 | 107.19 |
| CountGD (Amini-Naieni et al., 2024) | point-level | 12.14 | 47.51 | 14.76 | 120.42 |
| T2ICount (Qian et al., 2025) | point-level | 13.78 | 58.78 | 11.76 | 97.86 |
| VLPG (Zhai et al., 2025) | point-level | 16.05 | 53.49 | 17.60 | 97.66 |
| GCNet (Wang et al., 2024) | image-level | 19.50 | 63.13 | 17.83 | 102.89 |
| CLIP-Count* (Jiang et al., 2023) | image-level | 29.86 | 81.68 | 30.01 | 124.93 |
| T2ICount* (Qian et al., 2025) | image-level | 26.64 | 83.00 | 28.55 | 131.52 |
| MLLM-Zero | × | 38.92 | 119.26 | 38.19 | 145.42 |
| WS-COC-Base | image-level | 21.70 | 87.53 | 21.08 | 122.18 |
| WS-COC | image-level | **14.77** | **54.24** | **13.91** | **97.28** |

Table 2: Cross-dataset evaluation on CAPRK and PUCPR+.

| Method | Supervision | CARPK | | PUCPR+ | |
|---|---|---|---|---|---|
| | | MAE↓ | RMSE↓ | MAE↓ | RMSE↓ |
| CLIP-Count (Jiang et al., 2023) | point-level | 11.96 | 16.61 | - | - |
| VLCounter (Kang et al., 2024) | point-level | 6.46 | 8.68 | 48.94 | 69.08 |
| CountDiff (Hui et al., 2024) | point-level | 10.32 | 12.92 | - | - |
| VLPG (Zhai et al., 2025) | point-level | 10.14 | 13.79 | - | - |
| T2ICount (Qian et al., 2025) | point-level | 8.61 | 13.47 | - | - |
| CLIP-Count* (Jiang et al., 2023) | image-level | 24.90 | 29.12 | 63.28 | 86.63 |
| T2ICount* (Qian et al., 2025) | image-level | 19.42 | 23.48 | 58.43 | 77.24 |
| MLLM-Zero | × | 26.24 | 55.11 | 82.40 | 133.50 |
| WS-COC-Base | image-level | 14.06 | 27.60 | 60.40 | 84.64 |
| WS-COC | image-level | **10.39** | **15.83** | **42.30** | **54.06** |

comparable or even superior performance to some recent fully-supervised methods. For instance, WS-COC surpasses two very recent methods CountDiff (Hui et al., 2024) and VLPG (Zhai et al., 2025). It also achieves superior performance to weakly-supervised methods, *e.g.* , it decreases MAE by -3.92 and RMSE by -5.61 from GCNet (Wang et al., 2024) on the test set. Moreover, WS-COC significantly outperforms MLLM-Zero and the baseline model WS-COC-Base. It is worth noting that, as shown in Fig. 1(d), WS-COC substantially reduces the counting error (MAE) in dense scenes (more than 100 instances per image) from 149.69 (MLLM-Zero), and 82.44 (WS-COC-Base) to 54.37 (WS-COC). Some examples shown in Fig.3 further illustrate that WS-COC can produce accurate predictions.

Next, we follow previous methods (Jiang et al., 2023; Zhai et al., 2025; Hui et al., 2024) to conduct cross-dataset validation on the CARPK, PUCPR+ and ShanghaiTech datasets. As shown in Tab. 2 and Tab. 3, WS-COC achieves the best results on PUCPR+ and SHA, and exhibits competitive performance to most fully-supervised methods on CARPK and SHB. These results demonstrate the strong generalizability of our method in unknown scenarios.

Last, we compare the training and inference cost between WS-COC and some state-of-the-art methods (Jiang et al., 2023; Amini-Naieni et al., 2024; Qian et al., 2025) in Tab. 4, ensuring a fair comparison by using the same GPU (*i.e.* a NVIDIA L20). Although MLLMs are generally big, we adopt the LoRA-based finetuning paradigm, which helps significantly reduce the training cost. The training

Table 3: Cross-dataset evaluation on ShanghaiTech.

| Method | Supervision | SHA | | SHB | |
|---|---|---|---|---|---|
| | | MAE↓ | RMSE↓ | MAE↓ | RMSE↓ |
| RCC (Hobley & Prisacariu, 2023) | point-level | 240.1 | 366.9 | 66.6 | 104.8 |
| CLIP-Count (Jiang et al., 2023) | point-level | 192.6 | 308.4 | 45.7 | 77.4 |
| VLPG (Zhai et al., 2025) | point-level | 178.9 | 284.6 | 42.4 | 71.6 |
| GCNet (Wang et al., 2024) | image-level | 148.9 | 260.7 | 38.6 | **53.9** |
| CLIP-Count* (Jiang et al., 2023) | image-level | 237.4 | 355.5 | 52.3 | 73.8 |
| T2ICount* (Qian et al., 2025) | image-level | 222.2 | 367.6 | 48.9 | 76.9 |
| MLLM-Zero | × | 336.8 | 486.6 | 75.7 | 108.0 |
| WS-COC-Base | image-level | 190.4 | 343.0 | 43.6 | 71.8 |
| WS-COC | image-level | **128.9** | **232.9** | **34.2** | 57.0 |

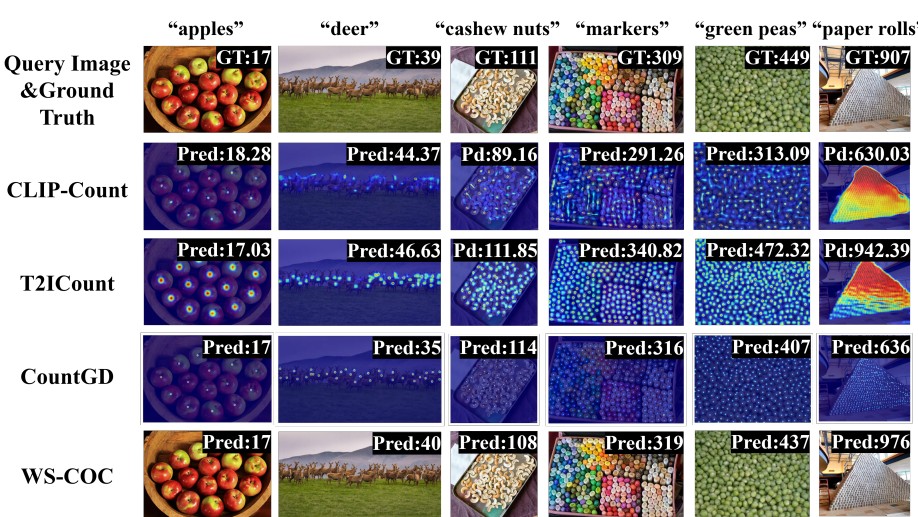

Figure 3: Visualization results among ground truth, CLIP-Count (Jiang et al., 2023), T2ICount (Qian et al., 2025), CountGD (Amini-Naieni et al., 2024), and WS-COC.

time of our WS-COC is only 3.44 hours, substantially lower than that of CountGD and T2ICount. Notice the primary computation cost of WS-COC is due to the inherent cost of the MLLM backbone (WS-COC-Base), rather than our proposed components. In practice, one may also adopt model compression techniques such as pruning and distillation to further reduce the computation overhead.

## 5.3 ABLATION STUDY

We conduct ablation study on the validation and test sets of FSC-147 dataset.

**Divide-and-discern dialogue tuning ($D^3T$).** In Tab. 5, we denote WS-COC w/o $D^3T$ as a variant in which $D^3T$ is removed from WS-COC. The results show a significant increase in MAE, rising by +3.36 on the validation set and +3.21 on the test set, indicating the effectiveness of the $D^3T$ strategy.

Next, we vary the stopping threshold $\delta$ in $D^3T$, from 0.1 to 0.3. As shown in Fig. 4(a), the default $\delta = 0.2$ performs the best on the validation set, and concurrently the best in testing.

Last, we apply $D^3T$ strategy in the testing stage named WS-COC w/ $D^3T$-T. As shown in Tab. 5, this variant performs notably worse, even inferior to WS-COC w/o $D^3T$. This is because, during training, we construct multi-round dialogue based on ground truth counts. However, at test time, any incorrect judgment in an intermediate round can mislead the MLLM into an entirely wrong direction, resulting in severely degraded performance.

**Compare-and-rank count optimization (CRCO).** In Tab. 5, we denote WS-COC w/o CRCO as a variant in which CRCO is removed. The results show a significant increase in MAE, rising by +2.62 on the validation set and +2.84 on the test set, indicating the effectiveness of this strategy.

Table 4: Comparison of training and inference cost.

| Method | Trainable Params (M) | Training Time (hours) | GFLOPs | Memory usage (GB) | FPS |
|---|---|---|---|---|---|
| CLIP-Count (Jiang et al., 2023) | 17.30 | 2.43 | 78.72 | 1.94 | 15.50 |
| CountGD (Amini-Naieni et al., 2024) | 37.00 | 11.73 | 3589.03 | 8.12 | 1.13 |
| T2ICount (Qian et al., 2025) | 908.42 | 23.84 | 1100.75 | 6.77 | 5.09 |
| WS-COC-Base | 339.94 | 1.84 | 1845.09 | 18.70 | 2.47 |
| WS-COC | 339.94 | 3.44 | 1845.09 | 18.70 | 2.16 |

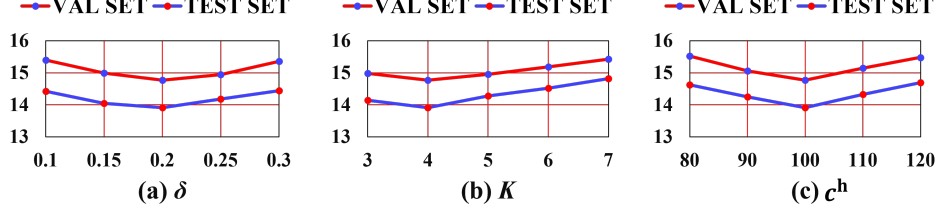

Figure 4: MAE on the FSC-147 validation and test sets when varying (a) the ratio of $\delta$ in $D^3T$, (b) the $K$ in CRCO, and (c) the dense threshold $c^h$ in GLCE.

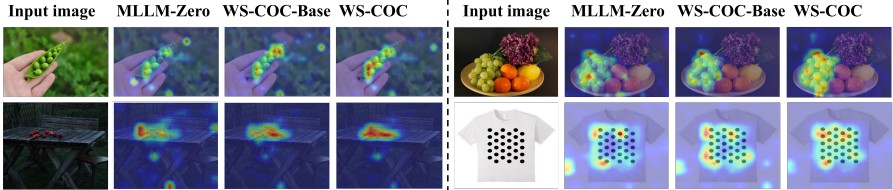

Figure 5: Attention map comparisons for MLLM-Zero, WS-COC-Base and WS-COC.

Next, we denote WS-COC w/ SCRCO as a variant, where we crop central regions of the image with sizes corresponding to $\frac{3}{4}$, $\frac{2}{4}$, and $\frac{1}{4}$ of the original dimensions. This is like in Matryoshka doll, similar to the idea used in CrowdCLIP (Liang et al., 2023). Together with the original image, these crops form an image set $\mathcal{I}$ of size $K = 4$, which is inherently ranked by the retained area. As shown in Tab. 5, this variant performs worse than our WS-COC and is even close to the WS-COC w/o CRCO. This is likely because such a ranking task (SCRCO) is much easier for MLLMs compared to the ranking task across different images (CRCO).

Furthermore, we vary $K$ in this strategy from 3 to 7. As shown in Fig. 4(b), we set $K = 4$ by default since it yields the best performance.

Last, we investigate the effectiveness of our proposed sampling strategy. We denote WS-COC w/ $\text{CRCO}_{rnd}$ as a variant, where images in $\mathcal{I}$ are obtained by randomly sampling from all images of a given category. As shown in Tab. 5, WS-COC w/ $\text{CRCO}_{rnd}$ produces certain performance gains compared to WS-COC w/o CRCO, yet the results are still inferior to that of our default setting, highlighting the benefit of ensuring a wide coverage of object counts during sample selection.

**Global-and-local counting enhancement (GLCE).** In Tab. 5, we denote WS-COC w/ GLCE ($c^g$/$c^l$) as a variant where GLCE is removed from WS-COC but only global prediction/aggregated local prediction is utilized. For the global-only variant ($c^g$), the results show a clear increase in MAE, rising by +1.87 on the validation set and +1.81 on the test set, indicating the effectiveness of GLCE; Similar observations can be found for the local-only variant ($c^l$).

Next, we denote WS-COC w/ GLCE($L = i$) as a variant where the input image is divided into a $i \times i$ grids for local enhancement. As shown in Tab. 5, our default $L = 2$ performs the best on the validation set, and concurrently also the best in testing.

We further conduct a detailed analysis of the global and local predicted counts in GLCE. We find that approximately 81.2% of global predictions $c^g$ underestimate object counts, while local aggregated predictions $c^l$, affected by the edge effect during aggregation over $\{c_k\}_{k=1}^{L^2}$, tend to overestimate

Table 5: Ablation study on the D$^3$T, CRCO, and GLCE.

| Method | VAL SET | | TEST SET | |
|---|---|---|---|---|
| | MAE↓ | RMSE↓ | MAE↓ | RMSE↓ |
| WS-COC w/o D$^3$T | 18.13 | 71.35 | 17.12 | 109.82 |
| WS-COC w/ D$^3$T-T | 28.90 | 80.58 | 37.07 | 123.70 |
| WS-COC w/o CRCO | 17.39 | 65.23 | 16.75 | 107.45 |
| WS-COC w/ SCRCO | 17.24 | 64.83 | 16.63 | 103.57 |
| WS-COC w/ CRCO$_{rnd}$ | 16.77 | 69.58 | 16.04 | 105.23 |
| WS-COC w/ GLCE ($c^g$) | 16.64 | 65.08 | 15.72 | 105.25 |
| WS-COC w/ GLCE ($c^l$) | 17.35 | 58.16 | 16.52 | 99.34 |
| WS-COC w/ GLCE($L=3$) | 15.28 | 57.64 | 14.34 | 99.93 |
| WS-COC w/ GLCE($L=4$) | 15.99 | 59.70 | 15.49 | 101.85 |
| WS-COC | **14.77** | **54.24** | **13.91** | **97.28** |

Table 6: Ablation study on different MLLM backbones of WS-COC.

| Method | VAL SET | | TEST SET | |
|---|---|---|---|---|
| | MAE↓ | RMSE↓ | MAE↓ | RMSE↓ |
| WS-COC (DeepSeek-VL2-3B) | 16.47 | 59.23 | 16.26 | 103.23 |
| WS-COC (DeepSeek-VL2-16B) | **14.45** | 56.71 | **13.67** | **95.51** |
| WS-COC (Qwen3-VL-4B) | 16.68 | 58.75 | 15.51 | 99.21 |
| WS-COC (Qwen3-VL-8B) | 15.18 | 55.90 | 14.13 | 97.23 |
| WS-COC (LLaVA-OneVersion-0.5B) | 17.05 | 60.87 | 17.32 | 105.89 |
| WS-COC (LLaVA-OneVersion-7B) | 14.77 | **54.24** | 13.91 | 97.28 |

object counts (about 79.4%). This complementary characteristic enables our solution to average them to improve the final counts.

Last, we vary the threshold $c^h$ in this strategy, from 80 to 120. As shown in Fig. 4, we set $c^h = 100$ by default, given it performs the best on both validation and test sets.

**Choice of MLLMs.** Tab. 6 presents the performance of WS-COC over different MLLM backbones and model sizes, including LLaVA-OneVision-7B (Li et al., 2024), LLaVA-OneVision-0.5B, Qwen3-VL-4B (Yang et al., 2025), Qwen3-VL-8B, DeepSeek-VL2-3B (Wu et al., 2024) and DeepSeek-VL2-16B. The results show that WS-COC generalizes well across different MLLM backbones: compared with our default LLaVA-OneVersion-7B, Qwen3-VL-8B with a similar model size yields comparable performance; on the other hand, the larger-scale DeepSeek-VL2-16B achieves better performance. Considering the efficiency and accuracy trade-off, we choose LLaVA-OneVision-7B by default.

**Visualization of attention maps.** To better understand the counting behavior of different models, we visualize the attention maps between the count-related text tokens and the image patch tokens for MLLM-Zero, WS-COC-Base and WS-COC. As shown in Fig. 5, the models progressively refine their focus towards objects that are to be counted, leading to more precise count predictions from MLLM-Zero to our WS-COC.

## 6    CONCLUSION

In this work, we present WS-COC, the first MLLM-driven weakly-supervised framework for class-agnostic object counting. The framework addresses the degradation of MLLM's counting performance in dense scenes through three strategies. First, the divide-and-discern dialogue tuning strategy reformulates count prediction as a series of count judgments, progressively breaking down the prediction range from coarse to fine through multi-round dialogue. Second, the compare-and-rank count optimization strategy trains the model to rank images with diverse object counts. Third, the global-and-local counting enhancement strategy combines the global prediction with aggregated local predictions from partitioned sub-images to mitigate counting bias in dense scenes during inference. Experiments across four benchmarks demonstrate the strong performance of WS-COC.

## 7 ACKNOWLEDGMENT

This work was supported by the Fundamental Research Funds for the Central Universities.

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

APPENDIX

This appendix provides 1) additional ablation study regarding the design choices of WS-COC components, the MLLM backbone selection, *etc.* and 2) additional analysis regarding cross-dataset evaluations and failure cases analysis.

## A    ADDITIONAL ABLATION STUDY

This section provides additional results, unless otherwise specified, on the validation and test sets of FSC-147.

**Multi-round dialogue in divide-and-discern dialogue tuning ($D^3T$).** Our proposed $D^3T$ strategy guides the MLLM to determine whether the object count falls within a specific range and progressively break down the range through multi-round dialogue in a curriculum manner, moving from easy to hard. We propose a variant that replaces the multi-round dialogue with a single-round dialogue, *i.e.*, the initial range ($[1, 2000]$) is directly divided into multiple sub-ranges with interval $\Delta$. For instance, for $\Delta = 200$, the sub-ranges are $[1, 200]$, $[201, 400]$, ..., $[1801, 2000]$. The MLLM is then queried to predict the correct sub-range for given images using the text instruction "Given the image, please determine into which range the number of [obj] falls: $\{[1, 200], [201, 400], \ldots, [1801, 2000]\}$.". We vary $\Delta$ and report the performance in Fig. A. The best performance is obtained with $\Delta = 300$, which yields only a slight improvement over WS-COC w/o $D^3T$ (see Tab. 5) and still remains clearly inferior to our $D^3T$. These results demonstrate the effectiveness of our multi-round dialogue in $D^3T$.

**Image sampling mechanism in compare-and-rank count optimization (CRCO).** In our CRCO strategy, the images used for count comparison are sampled from the same object category to ensure consistent visual semantics during ranking. We further propose a variant, denoted as WS-COC w/ $CRCO_{cross}$, where we follow the similar strategy in CRCO but sample images from all categories in the training set. As shown in Tab. A, this variant achieves certain gains over WS-COC w/o CRCO, but its performance is still inferior to our CRCO. This is because our CRCO preserves semantic consistency, which guides the model to focus on learning count difference among images. In contrast, cross-category sampling introduces semantic variance that degrades performance.

We also introduce another variant of CRCO that adopts a semi-cross-category sampling strategy. Specifically, we use GPT-5 to partition the object categories in the training set into semantically similar groups using the following prompt: "*Divide these categories into different groups, ensuring that categories within each group share semantic similarities: [categories]*". The grouping results are reported in Tab. B. We then sample images for CRCO from each group. We denote this variant by WS-COC w/ $CRCO_{semi-cross}$, and its performance is reported in Tab. A. We find that it performs slightly better than the fully cross-category sampling variant WS-COC w/ $CRCO_{cross}$, but it is still inferior to the single-category sampling mechanism used by default in CRCO. Although categories within the same group are semantically related, they still exhibit substantial appearance differences (*e.g.*, apples, pears, and oranges differ greatly in color), which can distract the MLLM from focusing on learning to count.

Last, we illustrate the model performance on each individual category in the test set of FSC-147, as shown in Fig. B, we observe that both WS-COC w/ $CRCO_{cross}$ and WS-COC w/ $CRCO_{semi-cross}$ underperform WS-COC in the vast majority of categories.

**Image partition mechanism in global-and-local counting enhancement (GLCE).** We have talked about the edge effect caused by local predictions in GLCE: the simple grid partition may split objects at the boundaries, potentially causing overestimation for objects located at the grid edges. In contrast, the global predictions tend to underestimate objects in dense scenes. Based on complementary nature of these two, we introduce GLCE to take advantage of both global and local predictions for accurate object counting. One way to alleviate this edge effect is to leverage segmentation masks generated by SAM (Kirillov et al., 2023) to construct partitions, hoping to ensure that each object is contained within only a single partition. We denote this variant by WS-COC w/ $GLCE_{SAM}$. In addition, LVLM-COUNT (Qharabagh et al., 2024) adopts a divide-and-conquer approach to alleviate this effect. It relies on additional pretrained models (GroundingDINO (Liu et al., 2024b) and SAM) to generate region proposals and object masks to divide the image for counting. This results in a

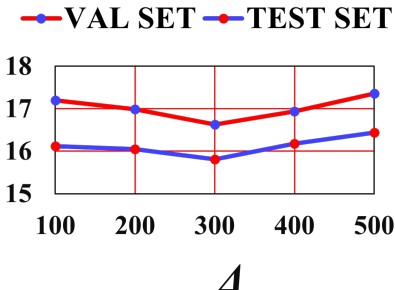

Figure A: MAE on the FSC-147 validation and test sets with different $\Delta$ in D$^3$T.

Table A: More ablation studies.

| Method | VAL SET | | TEST SET | |
|---|---|---|---|---|
| | MAE↓ | RMSE↓ | MAE↓ | RMSE↓ |
| WS-COC w/o CRCO | 17.39 | 65.23 | 16.75 | 107.45 |
| WS-COC w/ CRCO$_{cross}$ | 16.74 | 63.58 | 16.33 | 104.56 |
| WS-COC w/ CRCO$_{semi-cross}$ | 16.86 | 65.34 | 16.25 | 102.76 |
| WS-COC w/o GLCE | 16.64 | 65.08 | 15.42 | 105.25 |
| WS-COC w/ GLCE$_{SAM}$ | 16.59 | 70.18 | 15.34 | 103.14 |
| WS-COC w/ GLCE$_{LVLM}$ | 16.28 | 63.17 | 15.19 | 104.30 |
| MLLM-Zero | 38.92 | 119.26 | 38.19 | 145.42 |
| MLLM-Zero w/ GLCE | 33.49 | 102.83 | 32.65 | 128.70 |
| WS-COC-Base | 21.70 | 87.53 | 21.08 | 122.18 |
| WS-COC-Base w/ GLCE | 19.36 | 75.96 | 18.85 | 111.53 |
| WS-COC | **14.77** | **54.24** | **13.91** | **97.28** |

more complex pipeline and a strong dependency on the detection and segmentation quality, where errors can directly propagate to the final counting result. We denote this variant by WS-COC w/ GLCE$_{LVLM}$. As shown in Tab. A, both variants achieve trivial gains over WS-COC w/o GLCE but remain inferior to WS-COC. Our GLCE is overall simple, efficient, and effective.

Last, to examine the generality of GLCE, we integrater it into both MLLM-Zero and WS-COC-Base. As show in Tab. A, GLCE consistently improves the performance of these models, demonstrating its effectiveness as a general strategy.

**Dataset sensitivity of global-and-local counting enhancement (GLCE).** GLCE was proposed based on a consistent observation across datasets: MLLMs tend to underestimate object counts in dense scenes. This is owing to the nature that MLLMs have mostly been pre-trained on images with sparse crowds and small number of objects. Therefore, GLCE can be regarded as empirically dataset-insensitive. To verify this, we conduct detailed ablation study of GLCE on various datasets in Tab C, and we observe clear improvement on each of them. We further vary the partition grid size $L$ in GLCE. Although GLCE with $L = 3$ performs slightly better than $L = 2$ on SHA and PUCPR+, the overall gains are marginal and accompanied by a notable increase in computational cost. In contrast, GLCE with $L = 2$ consistently performs the best on FSC-147, SHB, and CARPK (results on FSC-147 are reported in Tab. 5 of the paper), achieving a favorable balance between efficiency and effectiveness.

## B    ADDITIONAL ANALYSIS

**Additional cross-dataset evaluation.** Beyond the cross-dataset evaluation reported in Tab. 2 and Tab. 3, we further conduct experiments on IOCfish5K (Sun et al., 2023) and PASCAL VOC

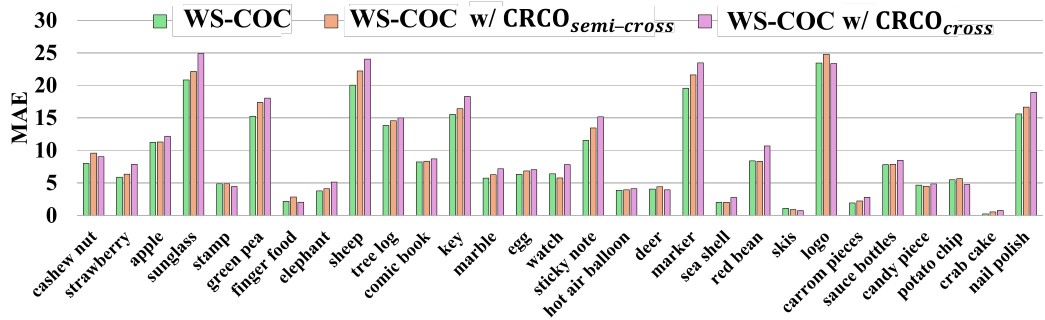

Figure B: Per-category MAE performance comparison of WS-COC under different sampling strategies in CRCO.

Table B: Grouping results of 89 categories in the training set of FSC-147.

| Group | Categories |
|---|---|
| Staple Foods and Pastries | buns, bread rolls, biscuits, cupcake tray, macarons, cupcakes, naan bread, croissants, baguette rolls, instant noodles, spring rolls |
| Fruits and Vegetables | oranges, potatoes, tomatoes, peppers, watermelon, bananas |
| Snacks and Prepared Food | ice cream, goldfish snack, chewing gum pieces, m&m pieces, meat skewers, onion rings, calamari rings |
| Beans and Grains | kidney beans, coffee beans, cereals, rice bags, nuts |
| Animals and Living Beings | pigeons, fishes, crows, geese, penguins, goats, swans, buffaloes, people, cows, bees, zebras, clams |
| Home and Building Materials | windows, mini blinds, roof tiles, bricks, cement bags, stairs, polka dot tiles, mosaic tiles, supermarket shelf |
| Transportation and Machinery | cars, boats, cranes |
| Tools and Consumables/Stationery | stapler pins, cartridges, matches, lighters, nails, screws, pencils, pens, crayons, coins, cassettes |
| Personal Items and Accessories/Daily Goods | jeans, shoes, lipstick, caps, kitchen towels, pearls, jade stones, gemstones, beads, candles, birthday candles, cotton balls, balls |
| Tableware, Containers and Miscellaneous Leisure | cups, plates, bowls, spoon, bottles, cans, boxes, alcohol bottles, chopstick, straws, go game |

2007 (Everingham et al., 2015) datasets, which involve visually ambiguous and heavily occluded objects. As shown in Tab. D, WS-COC achieves superior performance on both datasets, demonstrating its strong generalizability in handling visual ambiguity and occlusion. In particular, WS-COC significantly outperforms other fully-supervised methods on the PASCAL VOC 2007 dataset. This advantage may be attributed to the presence of similar data which have already been incorporated into the pre-training corpus of LLaVA-OneVision.

**Failure case analysis.** In Fig. C (a) and (b), we present some representative failure cases on FSC-147. The model may struggle in extremely dense scenes (see Fig. C (a)) and under severe occlusion (see Fig. C (b)), where individual instances are difficult to distinguish.

Table C: Performance comparison of GLCE variants on SHA, SHB, CARPK and PUCPR+.

| Method | SHA MAE↓ | SHA RMSE↓ | SHB MAE↓ | SHB RMSE↓ | CARPK MAE↓ | CARPK RMSE↓ | PUCPR+ MAE↓ | PUCPR+ RMSE↓ |
|---|---|---|---|---|---|---|---|---|
| WS-COC w/o GLCE | 147.1 | 263.9 | 41.7 | 64.3 | 13.78 | 20.99 | 52.76 | 67.32 |
| WS-COC w/ GLCE($L=2$) | 128.9 | 232.9 | **34.2** | **57.0** | **10.39** | 15.83 | 42.30 | 54.06 |
| WS-COC w/ GLCE($L=3$) | **127.4** | **227.7** | 38.5 | 57.7 | 11.14 | **15.64** | **41.64** | **53.95** |
| WS-COC w/ GLCE($L=4$) | 138.2 | 258.9 | 38.4 | 59.5 | 13.12 | 17.15 | 46.35 | 58.14 |

Table D: Cross-dataset evaluation on IOCfish5K and PASCAL VOC 2007.

| Method | Supervision | IOCfish5K MAE↓ | IOCfish5K RMSE↓ | PASCAL VOC 2007 MAE↓ | PASCAL VOC 2007 RMSE↓ |
|---|---|---|---|---|---|
| CLIP-Count (Jiang et al., 2023) | point-level | 82.1 | 155.2 | 12.5 | 32.7 |
| VLCounter (Kang et al., 2024) | point-level | 78.0 | 154.9 | 11.3 | 28.9 |
| CountGD (Amini-Naieni et al., 2024) | point-level | 67.4 | 144.8 | 8.7 | 21.7 |
| T2ICount (Qian et al., 2025) | point-level | 63.3 | 136.1 | 9.9 | 23.5 |
| WS-COC | image-level | 48.1 | 112.7 | 0.9 | 1.4 |

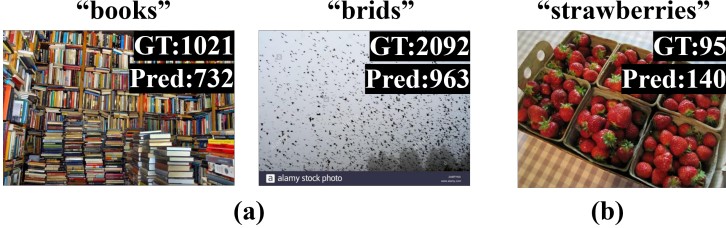

**"books"**  GT:1021  Pred:732

**"brids"**  GT:2092  Pred:963

**"strawberries"**  GT:95  Pred:140

(a)                          (b)

Figure C: (a) and (b) denote representative failure cases.

To further investigate the model's behavior in dense scenes, we respectively extract all "extremely dense scenes" (count > 300) from the validation and test sets of FSC-147. These two subsets are denoted as VAL EDS SET and TEST EDS SET in Tab. E. Next, we re-evaluate WS-COC and several representative methods (Jiang et al., 2023; Amini-Naieni et al., 2024; Qian et al., 2025) on them. We can observe that all methods perform poorly on these subsets, indicating that the difficulty is not specific to our method; rather, most existing methods (even fully-supervised ones) struggle with spatial understanding under such extremely dense conditions.

To shed light on the underlying cause, we analyze the correlation between object size and counting errors. When scenes contain hundreds of objects, each object normally occupies only a few pixels, making individual objects nearly indistinguishable from each other or from background textures. To quantitatively verify this observation, we compute the average object size for each image in the FSC-147 test set: in each image, three objects were originally annotated with bounding boxes for potential usage. Since in most images perspectives change insignificantly, we simply estimate the average object size of each image by averaging the three bounding box areas (in pixels). As shown in Fig. D, images with smaller average object sizes typically lead to larger counting errors.

Given above, we study whether finer-grained GLCE partitions (*i.e.* partitioning the input image into $3 \times 3$ or $4 \times 4$ grids) can improve the model performance on these two subsets. In Tab. E, we can see both variants (WS-COC w/ GLCE($L=3$) and WS-COC w/ GLCE($L=4$)) show some performance improvement, especially WS-COC w/ GLCE($L=3$). However, these improvements are only on these "extremely dense scenes" subsets, while on the full validation and test sets they do not add much benefit. Specifically, we implement another variant by selecting different grid sizes according to the predicted global count: for images with predicted global counts falling within [0,100), we directly use the global counts; for [100,300), we fuse the global count with local predictions using

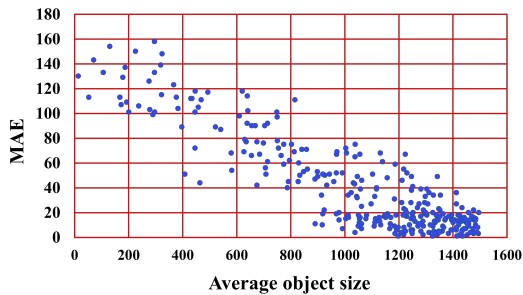

Figure D: Scatter plot of MAE versus average object size (in pixels) per image on the FSC-147 test set.

Table E: Quantitative evaluation in extremely dense scenes (more than 300 objects per image) of FSC-147.

| Method | Supervision | VAL EDS SET MAE↓ | RMSE↓ | TEST EDS SET MAE↓ | RMSE↓ |
|---|---|---|---|---|---|
| CLIP-Count (Jiang et al., 2023) | point-level | 219.59 | 348.80 | 227.14 | 684.11 |
| CountGD (Amini-Naieni et al., 2024) | point-level | 174.37 | 283.62 | 269.87 | 745.52 |
| T2ICount (Qian et al., 2025) | point-level | 205.07 | 315.80 | 212.35 | 666.94 |
| WS-COC | image-level | 185.97 | 297.95 | 226.96 | 647.36 |
| WS-COC w/ GLCE($L = 3$) | image-level | 180.28 | 291.10 | 221.48 | 638.62 |
| WS-COC w/ GLCE($L = 4$) | image-level | 183.14 | 293.06 | 237.33 | 649.25 |
| WS-COC w/ SynEds | image-level | 181.57 | 295.81 | 224.35 | 651.81 |

$2 \times 2$ grids ($L = 2$); for global counts above 300, we adopt $3 \times 3$ grids ($L = 3$). This variant achieves negligible performance gains compared to GLCE, *e.g.* -0.13 MAE on the full validation set of FSC-147. Since the performance benefit is very marginal while the design introduces much additional complexity, we choose to keep the current form of GLCE.

Besides, we explore whether data augmentation can alleviate this limitation. We manually concatenate multiple images of the same category from the FSC-147 training set to generate new images, each containing more than 300 objects. This process generates a total of 472 images. After integrating these augmented images into the original training set and retraining WS-COC, we observe very slight performance gains on both the VAL EDS SET and the TEST EDS SET (see WS-COC w/ SynEds versus WS-COC in Tab. E). However, this variant causes performance degradation compared with WS-COC on the full validation and test sets (*i.e.* +0.45 and +0.38 MAE respectively).

