# OpenReview forum: "Bootstrapping MLLM for Weakly‑Supervised  Class‑Agnostic Object Counting"
_ICLR.cc/2026/Conference — ICLR 2026 Poster_

### Official Review · Reviewer_XCqq · 2025-10-28

**Soundness:** 3
**Presentation:** 3
**Contribution:** 3
**Rating:** 6
**Confidence:** 4

**Summary:**

This paper proposes WS-COC, the first MLLM-driven weakly-supervised framework for class-agnostic object counting. Instead of directly fine-tuning MLLMs to predict object counts, WS-COC incorporates three key strategies: divide-and-discern dialogue tuning, compare-and-rank count optimization, and global-and-local counting enhancement. Extensive experiments are conducted on four benchmarks, showing that WS-COC matches or surpasses many fully-supervised methods while reducing annotation costs significantly.

**Strengths:**

1. This work innovatively proposes the first weakly supervised general category object counting framework WS-COC based on MLLM, filling the technical gap of "MLLM+weakly supervised+general category" object counting.
2. It designs three core strategies. The Divide-and-Discern Dialogue Tuning converts counting into multi-round range judgment to reduce learning difficulty; the Compare-and-Rank Count Optimization establishes the connection between visual features and counting through relative ranking; the Global-and-Local Counting Enhancement fuses multi-scale predictions to correct biases. These strategies optimize counting capabilities throughout the entire process from training to inference.
3. The experimental design is rigorous and comprehensive. It selects four representative benchmark datasets (FSC-147, CARPK, PUCPR+, and ShanghaiTech) that cover diverse scenarios and categories, ensuring the universality of results. Key implementation parameters such as MLLM backbone and LoRA settings are reported in detail to ensure experimental reproducibility.

**Weaknesses:**

1. Its performance is still insufficient in extremely dense scenes or under severe target occlusion. The model struggles to accurately distinguish individual instances, leading to cases of large counting errors.
2. In the Compare-and-Rank Count Optimization strategy, cross-category sampling introduces semantic differences and leads to performance degradation. Currently, image sampling can only be conducted within the same category, which limits the flexibility and application scope of the strategy.
3. The Global-and-Local Counting Enhancement strategy uses simple grid partitioning to split images into sub-images, which easily causes overestimation of local counts due to edge effects. Although this issue is alleviated by averaging with global counts, the fundamental problem caused by the partitioning method is not solved, leaving room for optimization.

**Questions:**

1. Regarding the failure cases in extremely dense scenes and severe occlusion (mentioned in the paper’s appendix), what specific characteristics of these scenes most strongly correlate with the model’s performance degradation? Is there a quantitative analysis to clarify the key factors limiting WS-COC’s performance in such scenarios?
2. In the Compare-and-Rank Count Optimization (CRCO) strategy, cross-category sampling reduces performance due to semantic variance. However, the paper does not specify whether the performance decline is consistent across all category pairs.
3. The Global-and-Local Counting Enhancement (GLCE) strategy uses a fixed threshold to distinguish dense and sparse scenes. How was this threshold determined?
4. Extend CRCO to support semi-cross-category sampling: Instead of restricting sampling to the same category, test a "semi-cross-category" sampling strategy. For example, group semantically similar categories (e.g., "apple", "pear", "orange" as a "fruit group") and sample images within each group.
5. Optimize the dense-scene threshold c^h adaptively.

---

> ### Author Response · Authors · 2025-11-22
> **Response to Weakness 1 and 2 of Reviewer XCqq**
>
> > **Weakness 1:**
> >
> >Its performance is still insufficient in extremely dense scenes or under severe target occlusion. The model struggles to accurately distinguish individual instances, leading to cases of large counting errors.
>
> **Response:**
>
> We appreciate your comment. Extremely dense scenes or severe occlusions remain challenging for many existing counting approaches because objects become very small and heavily occluded in such scenarios. As shown in the table below, when evaluating scenes with more than 300 objects on FSC-147 dataset, existing methods (even fully-supervised ones) exhibit large counting errors. To mitigate this limitation, we introduce strategies at both the training and inference stages. During training, we propose the Divide-and-Discern Dialogue Tuning strategy. Specifically, instead of regressing the exact count in one shot (often lead to large counting errors), we guide the model to iteratively determine whether the count falls within specific ranges through multi-round dialogue. By progressively breaking down the range from coarse to fine, the model is able to learn to count in a curriculum manner, moving from easy to hard.
> During inference, we observe that the model tends to underestimate object counts in dense scenes. Therefore, we propose Global-and-Local Counting Enhancement strategy. Specifically, we partition the images with dense scenes into smaller sub-images and query the MLLM on each sub-image independently. The local predictions are aggregated and combined with the global prediction to produce the final output. Experiments in the table below show that WS-COC achieves competitive performance with other fully-supervised methods, indicating that the proposed strategies help alleviate this limitation to a noticeable extent. In future work, we plan to explore more effective data augmentation and finer-grained region reasoning strategies to further address this challenge.
>
> | Method | \| Supervision | \| VAL EDS SET MAE  | VAL EDS SET RMSE  | \| TEST EDS SET MAE | TEST EDS SET RMSE  |
> |--------|----------------|----------------------|--------------------|-----------------------|---------------------|
> | CLIP-Count (Jiang et al., ACM-MM2023) | \| point-level | \| 219.59 | 348.80 | \| 227.14 | 684.11 |
> | CountGD (Amini-Naieni et al., NeurIPS2024) | \| point-level | \| 174.37 | 283.62 | \| 269.87 | 745.52 |
> | T2ICount (Qian et al., CVPR2025) | \| point-level | \| 205.07 | 315.80 | \| 212.35 | 666.94 |
> | WS-COC | \| image-level | \| 185.97 | 297.95 | \| 226.96 | 647.36 |
>
> > **Weakness 2:**
> >
> >In the Compare-and-Rank Count Optimization (CRCO) strategy, cross-category sampling reduces performance due to semantic variance. However, the paper does not specify whether the performance decline is consistent across all category pairs.
>
> **Response:**
>
> Thanks and we have done this. As discussed in the Appendix, WS-COC w/ CRCOcross  indicates a variant in which we follow the similar strategy in CRCO but sample images from all categories in the training set. This variant performs worse than the single-category sampling mechanism used in our WS-COC. We further analyze the model performance on each individual category in the test set of FSC-147, as shown in the [figure](https://anonymous.4open.science/r/additional-materials-6BD8/Fig4.png) (please click to view), we observe that WS-COC w/  CRCOcross underperforms our WS-COC in the vast majority of categories (25 out of 29). This is because cross-category sampling introduces semantic variance, which tends to make the MLLM focus on distinguishing different object categories rather than learning the count differences among images.

---

> ### Author Response · Authors · 2025-11-22
> **Response to Weakness 3 and Question 1, 2, 3, and 5 of Reviewer XCqq**
>
> > **Weakness 3:**
> >
> > The Global-and-Local Counting Enhancement strategy uses simple grid partitioning to split images into sub-images, which easily causes overestimation of local counts due to edge effects. Although this issue is alleviated by averaging with global counts, the fundamental problem caused by the partitioning method is not solved, leaving room for optimization.
>
> **Response:**
>
> Thanks. We examine widely used datasets and consistently observe a clear trend of overestimation when summing local predictions on them, primarily due to the inherent edge effects introduced by grid partitioning. Our proposed Global-and-Local Counting Enhancement (GLCE) strategy effectively alleviate this bias by aggregating global and local predictions and therefore remains suitable and robust across different datasets. In Appendix Tab.A, we have provided a variant of GLCE in which we leverage segmentation masks generated by SAM to construct partitions to alleviate these edge effects,  and the final count is obtained by aggregating the local partition predictions. However, this variant remains inferior to our original GLCE because segmentation errors can directly propagate to the final counting results.
>
> > **Question 1:**
> >
> > Regarding the failure cases in extremely dense scenes and severe occlusion (mentioned in the paper’s appendix), what specific characteristics of these scenes most strongly correlate with the model’s performance degradation? Is there a quantitative analysis to clarify the key factors limiting WS-COC’s performance in such scenarios?
>
> **Response:**
>
> Thanks. Our analysis shows that the key factor strongly correlated with performance degradation is the very small object size in such cases. When scenes contain hundreds of objects, each object normally occupies only a few pixels, making individual objects nearly indistinguishable from each other or from background textures, which fundamentally limits the model’s counting performance. To quantitatively verify this observation, we compute the average object size for each image in the FSC-147 test set: in each image, three objects were originally annotated with bounding boxes for potential usage.
> Since in most images perspectives change insignificantly, we simply estimate the average object size of each image by averaging the three bounding box areas (in pixels). As shown in the [figure](https://anonymous.4open.science/r/additional-materials-6BD8/Fig3.png) (please click to view), images with smaller average object sizes typically lead to larger counting errors.
>
> > **Question 2:**
> >
> >In the Compare-and-Rank Count Optimization (CRCO) strategy, cross-category sampling reduces performance due to semantic variance. However, the paper does not specify whether the performance decline is consistent across all category pairs.
>
> **Response:**
>
> Thanks. As shown in the [figure](https://anonymous.4open.science/r/additional-materials-6BD8/Fig4.png) (please click to view), we observe that cross-category sampling variant (WS-COC w/ CRCOcross) performs inferior to our WS-COC in the vast majority of categories (25 out of 29).
>
> > **Question 3 and 5:**
> >
> >The Global-and-Local Counting Enhancement (GLCE) strategy uses a fixed threshold to distinguish dense and sparse scenes. How was this threshold determined?
> >
> >Optimize the dense-scene threshold $c^h$  adaptively.
>
>
> **Response:**
>
> Thanks. The threshold $c^h$ in the GLCE strategy is determined based on the model performance on the validation set of FSC-147 (shown in Fig.4(c) in the paper).

---

> ### Author Response · Authors · 2025-11-22
> **Response to Question 4 of Reviewer XCqq**
>
> > **Question 4:**
> >
> > Extend CRCO to support semi-cross-category sampling: Instead of restricting sampling to the same category, test a "semi-cross-category" sampling strategy. For example, group semantically similar categories (e.g., "apple", "pear", "orange" as a "fruit group") and sample images within each group.
>
> **Response:**
>
> Thanks. Following your advice, we introduce another variant of CRCO that adopts a semi-cross-category sampling strategy. Specifically, we use GPT-5 to partition the object categories in the training set into semantically similar groups using the following prompt:  "Divide these categories into different groups, ensuring that categories within each group share semantic similarities: [categories]".
>
> The grouping results are reported in Table II. We then sample images for CRCO from each group.
>
> We denote this variant as WS-COC w/ CRCOsemi-cross, and its performance is reported in Table I. We find that it performs slightly better than the fully cross-category sampling variant, but it is still inferior to the single-category sampling mechanism used by default in CRCO.
>
> Although categories within the same group are semantically related, they still exhibit substantial appearance differences (e.g., apples, pears, and oranges differ greatly in color), which can distract the MLLM from focusing on learning to count.
>
> As shown in the [figure](https://anonymous.4open.science/r/additional-materials-6BD8/Fig4.png) (please click to view), this variant underperforms WS-COC in approximately 24 out of 29 categories.
>
> **Table I**
> | Method | \| VAL SET MAE  | VAL SET RMSE  | \| TEST SET MAE  | TEST SET RMSE  |
> |--------|------------------|----------------|-------------------|-----------------|
> | WS-COC w/ CRCOcross | \| 17.24 | 64.83 | \| 16.63 | 103.57 |
> | WS-COC w/ CRCOsemi-cross| \| 16.86 | 65.34 | \| 16.25 | 102.76 |
> | **WS-COC** | \| **14.77** | **54.24** | \| **13.91** | **97.28** |
>
>
> **Table II**
>
> | Group | \| Categories |
> |-------|---------------|
> | Staple Foods and Pastries | \| buns, bread rolls, biscuits, cupcake tray, macarons, cupcakes, naan bread, croissants, baguette rolls, instant noodles, spring rolls |
> | Fruits and Vegetables | \| oranges, potatoes, tomatoes, peppers, watermelon, bananas |
> | Snacks and Prepared Food | \| ice cream, goldfish snack, chewing gum pieces, m&m pieces, meat skewers, onion rings, calamari rings |
> | Beans and Grains | \| kidney beans, coffee beans, cereals, rice bags, nuts |
> | Animals and Living Beings | \| pigeons, fishes, crows, geese, penguins, goats, swans, buffaloes, people, cows, bees, zebras, clams |
> | Home and Building Materials | \| windows, mini blinds, roof tiles, bricks, cement bags, stairs, polka dot tiles, mosaic tiles, supermarket shelf |
> | Transportation and Machinery | \| cars, boats, cranes |
> | Tools and Consumables / Stationery | \| stapler pins, cartridges, matches, lighters, nails, screws, pencils, pens, crayons, coins, cassettes |
> | Personal Items and Accessories / Daily Goods | \| jeans, shoes, lipstick, caps, kitchen towels, pearls, jade stones, gemstones, beads, candles, birthday candles, cotton balls, balls |
> | Tableware, Containers and Miscellaneous Leisure | \| cups, plates, bowls, spoon, bottles, cans, boxes, alcohol bottles, chopstick, straws, go game |

---

### Official Review · Reviewer_QMGG · 2025-10-30

**Soundness:** 3
**Presentation:** 3
**Contribution:** 3
**Rating:** 8
**Confidence:** 4

**Summary:**

This paper introduces the use of Multimodal Large Language Models (MLLMs) for weakly-supervised, class-agnostic object counting. To enhance MLLMs’ performance on dense object scenes, the authors propose two training strategies and one inference strategy. Experimental results demonstrate that the proposed approach achieves state-of-the-art performance among weakly-supervised models and is only slightly inferior to the best fully-supervised counterparts. Ablation studies further validate the effectiveness of the proposed strategies.

**Strengths:**

- The paper is one of the first attempts to explore the use of MLLMs for weakly-supervised, class-agnostic object counting. The results achieved by the proposed method are competitive with those of fully-supervised approaches, which is a notable contribution to the field.
- The proposed training and inference strategies are simple yet effective, and the experiments are meticulously structured.

**Weaknesses:**

- **The paper lacks analysis of inference speed and computational cost**. Since MLLMs are computationally expensive, it is important to quantify inference time, memory usage, and FLOPs, as these factors are critical for real-world deployment.
- **Limited mechanism interpretability**. Although the proposed method is empirically effective, the underlying mechanisms that drive the improvement remain unclear. Visualizations of model attention or response patterns (e.g., Grad-CAM on the vision-language alignment) may help elucidate this.
- **The paper lacks qualitative comparisons with other object counting models**. The visual results only compare different variants within the MLLM family, but omit comparisons with existing leading density-map based or detection-based weakly- or fully-supervised baselines. Including such comparisons could help clarify the strengths of the proposed method relative to other approaches.

**Questions:**

- MLLMs are known to incur high inference costs, making them unsuitable for deployment on resource-constrained devices. What potential application scenarios or deployment strategies do the authors envision for MLLM-based counting models?
- Could the authors provide visual or attention-based analyses to better understand how the model perceives and counts objects? This would help clarify how the model interprets scenes and focuses on visual cues during the counting process.

---

> ### Author Response · Authors · 2025-11-22
> **Response to Weakness 1,2 and Question 2 of Reviewer QMGG**
>
> > **Weakness 1:**
> > **The paper lacks analysis of inference speed and computational cost.** Since MLLMs are computationally expensive, it is important to quantify inference time, memory usage, and FLOPs, as these factors are critical for real-world deployment.
>
>  **Response:**
> We appreciate your comment. Indeed, MLLMs are generally big, as the first of its kind to leverage MLLM for object counting, we can not deny it.
> Nonetheless, instead of adopting full finetuning, we adopt the LoRA-based finetuning paradigm, this helps significantly reduce the training cost. Below in the table, we compare the inference speed, FLOPs and memory usage between WS-COC and several state-of-the-art  methods, ensuring a fair comparison by using the same GPU (i.e. a NVIDIA L20). We can see that WS-COC achieves faster inference speed (fps) and lower FLOPs than CountGD, yet it is slower than T2ICount and CLIP-Count. This is primarily due to the inherent computational cost of the MLLM backbone (WS-COC-Base), rather than our proposed components.
> In practical applications, one may select MLLMs of appropriate sizes according to deployment needs, or adopt model compression techniques such as pruning and distillation to reduce the throughput burden of large models.
>
>
> | Method | \| fps ↑ | \| GFLOPs ↓ | \| GPU Memory Usage (MB) ↓ |
> |--------|----------|----------|----------------------------|
> | CLIP-Count (Jiang et al., ACM-MM2023) | \| 15.50 | \|78.72   | \| 1940.82  |
> | CountGD (Amini-Naieni et al., NeurIPS2024) | \| 1.13  | \|3589.03 | \| 8124.33  |
> | T2ICount (Qian et al., CVPR2025) | \| 5.09  | \|1100.75 | \| 6766.96  |
> | WS-COC-Base | \| 2.47 | \|1845.09 |  \|18698.43 |
> | WS-COC | \| 2.16 | \|1845.09 |  \|18698.43 |
>
> > **Weakness 2 and Question 2 :**
> > **Limited mechanism interpretability.**  Although the proposed method is empirically effective, the underlying mechanisms that drive the improvement remain unclear. Visualizations of model attention or response patterns (e.g., Grad-CAM on the vision-language alignment) may help elucidate this.
> >
> >Could the authors provide visual or attention-based analyses to better understand how the model perceives and counts objects? This would help clarify how the model interprets scenes and focuses on visual cues during the counting process.
>
>  **Response:**
> Thanks. We visualize the attention maps between the count-related text tokens and the image patch tokens inside the MLLM. Specifically, during auto-regressive text generation process, count-related text tokens attend to vision tokens through the MLLM’s self-attention mechanism. We extract the corresponding attention weights from the last six layers of the MLLM and average them to highlight the image regions that the model focuses on when generating count-related tokens in the output text.
> As shown in the [figure](https://anonymous.4open.science/r/additional-materials-6BD8/Fig1.png)  (please click to view), the attention maps indicate that the model indeed focuses on object regions and therefore outputs precise count predictions.

---

> > ### Comment · Reviewer_QMGG · 2025-11-25
> >
> > Thanks for the author's reply. The visualizations of the MLLM’s internal attention maps are clear and helpful. However, the analysis could be substantially strengthened by providing comparative attention visualizations across MLLM-Zero, WS-COC-Base, and the final WS-COC model. Such a comparison may more clearly illustrate how each component contributes to progressively refining the model’s focus on the objects to be counted.

---

> > > ### Author Response · Authors · 2025-11-25
> > > **Response to Reviewer QMGG**
> > >
> > > Thank you very much for your recognition. As suggested, we provide attention map comparisons for MLLM-Zero, WS-COC-Base, and our WS-COC in the [figure](https://anonymous.4open.science/r/additional-materials-6BD8/Fig6.png) (please click to view). We can see the model’s focus has been progressively refined towards objects that are to be counted.

---

> ### Author Response · Authors · 2025-11-22
> **Response to Weakness 3 and Question 1 of Reviewer QMGG**
>
> > **Weakness 3:**
> >
> >**The paper lacks qualitative comparisons with other object counting models.**  The visual results only compare different variants within the MLLM family, but omit comparisons with existing leading density-map based or detection-based weakly- or fully-supervised baselines. Including such comparisons could help clarify the strengths of the proposed method relative to other approaches.
>
> **Response:**
>
> Thanks. In the [figure](https://anonymous.4open.science/r/additional-materials-6BD8/Fig2.png) (please click to view), we provide qualitative comparisons with existing methods, including two density-map based methods CLIP-Count (Jiang et al., ACM-MM2023) and T2ICount (Qian et al., CVPR2025), and a detection-based method CountGD (Amini-Naieni et al., NeurIPS2024). All these methods are fully-supervised approaches, while our WS-COC is weakly-supervised. We can see our WS-COC produces more accurate predictions, particularly in challenging dense scenarios.
>
>
> > **Question 1:**
> >
> >MLLMs are known to incur high inference costs, making them unsuitable for deployment on resource-constrained devices. What potential application scenarios or deployment strategies do the authors envision for MLLM-based counting models?
>
> **Response:**
>
> Thanks. We acknowledge that our MLLM-based model incur relatively high inference costs. However, there are many practical application scenarios where such models remain highly valuable: 1) Cloud-based counting services for crowd and vehicle monitoring at traffic intersections, where inference can be executed on powerful cloud servers rather than on edge devices. 2) Aerial image analysis, such as counting trees, houses, or other large-scale objects in remote sensing imagery, where processing is typically performed on offline servers with sufficient computational resources. In addition, we believe that deployment strategies such as model distillation, model quantization, and model pruning can further mitigate the inference cost. Last, we emphasize the impressive performance obtained by our MLLM-driven weakly-supervised object counting framework, which surpasses many fully-supervised methods, enabling a wider usage of it in solving the open-world counting problem.

---

### Official Review · Reviewer_GPEv · 2025-11-01

**Soundness:** 3
**Presentation:** 3
**Contribution:** 3
**Rating:** 6
**Confidence:** 5

**Summary:**

This paper introduces WS-COC, a novel framework for weakly-supervised, class-agnostic object counting. This is the first work, to my knowledge, to successfully bootstrap a Multimodal Large Language Model (MLLM) for this task using only image-level count supervision. The core problem is that MLLMs, while capable of basic counting, fail significantly in dense scenes.

**Strengths:**

The paper tackles a highly practical and important problem: class-agnostic counting without expensive point-level annotations. The approach of "bootstrapping" a generative MLLM (LLaVA) for this counting task, rather than building a discriminative VLM (like CLIP) with a specialized counting head, is a novel and promising direction.

The experimental validation is thorough. WS-COC's performance on the primary benchmark, FSC-147, is excellent, significantly reducing the MAE in dense scenes compared to baselines (Fig. 1(d)).

**Weaknesses:**

The paper's CRCO strategy is well-motivated as a way to handle the modality gap using relative ranking. However, the related work section overlooks prior art from the VLM (Vision-Language Model) domain that has explored similar concepts. A discussion in Section 2.1 or 2.2 situating WS-COC's CRCO strategy against VLM-based ranking methods like CrowdCLIP would significantly strengthen the paper's contribution and provide clearer context on its novelty.


The failure case analysis shows the model still struggles with "extremely dense scenes" (e.g., "books" GT: 1021, Pred: 732). Is this a fundamental limitation of the MLLM's spatial understanding and resolution, or could this be further mitigated by, for instance, a finer-grained GLCE partitioning (e.g., $L=3$ or $L=4$, which Table 4 shows performed worse) or a more sophisticated data augmentation strategy for dense scenes?

**Questions:**

see weakness

---

> ### Author Response · Authors · 2025-11-22
> **Response to Weakness 1 of Reviewer GPEv**
>
> > **Weakness 1:**
> > The paper's CRCO strategy is well-motivated as a way to handle the modality gap using relative ranking. However, the related work section overlooks prior art from the VLM (Vision-Language Model) domain that has explored similar concepts. A discussion in Section 2.1 or 2.2 situating WS-COC's CRCO strategy against VLM-based ranking methods like CrowdCLIP would significantly strengthen the paper's contribution and provide clearer context on its novelty.
>
> **Response:**
>
> We appreciate your comment. Our CRCO strategy clearly differs from the ranking mechanisms used in VLM-based methods like CrowdCLIP in several aspects:  1) They normally generate an image set by cropping the given image into sub-images of different sizes, assuming that larger crops naturally contain more objects to form a rank. In contrast, WS-COC constructs the image set by sampling different images with varying numbers of objects within the same category.  Regarding the former, we have already implemented a similar “Matryoshka-doll-style” ranking strategy in our ablation study, denoted by WS-COC w/ SCRCO in Tab.4 of the paper. This variant performs worse than our WS-COC and is even close to the WS-COC w/o CRCO. This is likely because such a ranking task (SCRCO) is much easier for MLLMs compared to the ranking task across different images (CRCO), thereby limiting its effectiveness in enhancing MLLM's counting ability.  2) VLM-based ranking methods optimize a contrastive objective to align image features with text features of corresponding descriptions, as VLMs are fundamentally discriminative models that rely on feature alignment to perform ranking. In contrast, we are the first to leverage a generative model (MLLM) for the object counting task: we directly query the MLLM to auto-regressively predict the ranks using the language modeling loss.

---

> ### Author Response · Authors · 2025-11-22
> **Response to Weakness 2 of Reviewer GPEv**
>
> > **Weakness 2:**
> >The failure case analysis shows the model still struggles with ''extremely dense scenes" (e.g., "books" GT: 1021, Pred: 732). Is this a fundamental limitation of the MLLM's spatial understanding and resolution, or could this be further mitigated by, for instance, a finer-grained GLCE partitioning (e.g., L=3 or L=4, which Table 4 shows performed worse) or a more sophisticated data augmentation strategy for dense scenes?
>
> **Response:**
>
> Thanks. We respectively extract all “extremely dense scenes” (count $>$ 300) from the validation and test sets of FSC-147. These two subsets are denoted as VAL EDS SET and TEST EDS SET in the table below. Next, we re-evaluate WS-COC and several representative methods on them. We observe that all methods perform unsatisfactory on these subsets, indicating that the difficulty is not specific to MLLMs; rather, most existing methods (even fully-supervised ones) struggle under such extremely dense conditions.
>
> We further evaluate finer-grained GLCE partitions (i.e. partitioning the input image into   $3\times 3 $ or $4\times 4$ grids) on these two subsets. We can see both variants (WS-COC w/ GLCE(L = 3) and WS-COC w/ GLCE(L = 4)) show some performance improvement, especially WS-COC w/ GLCE(L = 3). However, these improvements are only  on the “extremely dense scenes” subsets, while on the full validation and test sets they do not add much benefit.
>
> Specifically, we implement another variant by selecting different grid sizes according to the predicted global count: for images with predicted global counts falling within [0,100), we directly use the global counts; for [100,300), we fuse the global count with local predictions using $2\times 2$ grids (L=2); for global counts above 300, we adopt $3\times3$ grids (L=3). This variant achieves negligible performance gains compared to the default GLCE, e.g. -0.13 MAE on the entire validation set of FSC-147. Since the performance benefit is very marginal while the design introduces much additional complexity, we choose to keep the original form of GLCE.
>
> Moreover, we follow your idea to explore whether data augmentation can alleviate this limitation. In particular, we manually concatenate multiple images of the same category from the FSC-147 training set to generate new images, each containing more than 300 objects. This process generates a total of 472 images. After integrating these augmented images into the original training set and retraining WS-COC, we observe slight performance gains on both the VAL EDS SET and the TEST EDS SET (see WS-COC w/ SynEds v.s. WS-COC in the table below). However, this variant causes performance degradation compared with WS-COC on the full validation and test sets (i.e.  +0.45 and +0.38 MAE respectively), therefore we do not adopt it.
>
>
> | Method | \| Supervision | \| VAL EDS SET MAE  | VAL EDS SET RMSE  | \| TEST EDS SET MAE  | TEST EDS SET RMSE  |
> |--------|----------------|----------------------|--------------------|-----------------------|---------------------|
> | CLIP-Count (Jiang et al., ACM-MM2023) | \| point-level | \| 219.59 | 348.80 | \| 227.14 | 684.11 |
> | CountGD (Amini-Naieni et al., NeurIPS2024) | \| point-level | \| 174.37 | 283.62 | \| 269.87 | 745.52 |
> | T2ICount (Qian et al., CVPR2025) | \| point-level | \| 205.07 | 315.80 | \| 212.35 | 666.94 |
> | WS-COC | \| image-level | \| 185.97 | 297.95 | \| 226.96 | 647.36 |
> | WS-COC w/ GLCE (L=3) | \| image-level | \| 180.28 | 291.10 | \| 221.48 | 638.62 |
> | WS-COC w/ GLCE (L=4) | \| image-level | \| 183.14 | 293.06 | \| 237.33 | 649.25 |
> | WS-COC w/ SynEds | \| image-level | \| 181.57 | 295.81 | \| 224.35 | 651.81 |

---

### Official Review · Reviewer_3dgp · 2025-11-03

**Soundness:** 3
**Presentation:** 3
**Contribution:** 3
**Rating:** 6
**Confidence:** 3

**Summary:**

This paper proposes WS-COC, a weakly-supervised class-agnostic object counting framework that bootstraps multimodal large language models for object counting using only image-level supervision. Instead of directly fine-tuning MLLMs for regression, the authors introduce three strategies: (1) Divide-and-Discern Dialogue Tuning, which reformulates counting as iterative range judgment through multi-round dialogue; (2) Compare-and-Rank Count Optimization, which trains the MLLM to rank multiple images by their object counts; and (3) Global-and-Local Counting Enhancement, which fuses global and local predictions at inference to better handle dense scenes. Experiments show that WS-COC presents good performance while using only weak supervision.

**Strengths:**

- The paper is clearly written and well-structured, with effective figures that help illustrate the intuition behind each proposed component.
- The three proposed strategies are conceptually simple yet effective, with clear motivations.
- The method achieves better results compared to other methods under image-level supervision.

**Weaknesses:**

- The paper does not discuss computational efficiency. Since the framework relies on large multimodal language models and multi-round dialogues, a comparison of training and inference cost against other weakly- and fully-supervised baselines would help clarify its practical scalability and deployment feasibility.
- The experiments are conducted on general object counting datasets; however, the model’s ability to generalize to other domains remains unclear. For example, it would be valuable to investigate how the proposed framework performs on indiscernible object counting tasks where objects are visually ambiguous or heavily occluded. The related references are listed below.
- Missing related work. Several recent works on object counting are relevant and should be discussed:
   - A. Distribution Matching for Crowd Counting
   - B. Indiscernible object counting in underwater scenes
   - C. Counting Everyday Objects in Everyday Scenes

**Questions:**

Please see the points listed in weakness section.

---

> ### Author Response · Authors · 2025-11-22
> **Response to Weakness 1 of Reviewer 3dgp**
>
> >**Weakness 1:**
> >The paper does not discuss computational efficiency. Since the framework relies on large multimodal language models and multi-round dialogues, a comparison of training and inference cost against other weakly- and fully-supervised baselines would help clarify its practical scalability and deployment feasibility.
>
> **Response:**
> We appreciate your comment. Indeed, MLLMs are generally big; as the first of its kind to leverage MLLM for object counting, we cannot deny it. Nevertheless, instead of adopting full finetuning, we adopt the LoRA-based finetuning paradigm, which helps significantly reduce the training cost. Below in the table we compare the training and inference cost between WS-COC and some state-of-the-art methods, ensuring a fair comparison by using the same GPU (i.e. a NVIDIA L20). We can see WS-COC's training time is longer than CLIP-Count (Jiang et al., ACM-MM2023) but significantly shorter than T2ICount (Qian et al., CVPR2025) and CountGD (Amini-Naieni et al., NeurIPS2024), while its inference speed is slower than CLIP-Count and T2ICount but faster than CountGD. WS-COC's computational cost is primarily due to the inherent cost of the MLLM backbone (WS-COC-Base), rather than our proposed components. In fact, our proposed divide-and-discern dialogue tuning strategy adds only negligible computation.
>
>
> | Method |\| Trainable Params (M) ↓ | \|Training Time (hours) ↓ | \|fps ↑ |
> |--------|-------------------------|--------------------------|-------|
> | CLIP-Count (Jiang et al., ACM-MM2023) | \|17.30 | \|2.43 | \|15.50 |
> | CountGD (Amini-Naieni et al., NeurIPS2024) | \|37.00 | \|11.73 | \|1.13 |
> | T2ICount (Qian et al., CVPR2025) | \|908.42 | \|23.84 | \|5.09 |
> | WS-COC-Base | \|339.94 |\|1.84 | \|2.47 |
> | WS-COC w/o D³T |\|339.94 | \|2.56 | \|2.16 |
> | WS-COC |\|339.94 |\|3.44 | \|2.16 |

---

> ### Author Response · Authors · 2025-11-22
> **Response to Weakness 2 of Reviewer 3dgp**
>
> > **Weakness 2:**
> > The experiments are conducted on general object counting datasets; however, the model’s ability to generalize to other domains remains unclear. For example, it would be valuable to investigate how the proposed framework performs on indiscernible object counting tasks where objects are visually ambiguous or heavily occluded. The related references are listed below.
> >
> > Missing related work. Several recent works on object counting are relevant and should be discussed:
> >
> > A. Distribution Matching for Crowd Counting
> > B. Indiscernible object counting in underwater scenes
> > C. Counting Everyday Objects in Everyday Scenes
>
>  **Response:**
> Thanks. Our experiments already follow previous works by evaluating WS-COC across different domains, including crowd counting (ShanghaiTech) in Tab.3 of the paper and vehicle counting (CARPK, PUCPR+) in Tab.2 of the paper. Especially,  crowds in ShanghaiTech SHA are already quite dense and heavily occluded (the average crowd count in SHA is 501.40 per image); notwithstanding, our method achieves an MAE of 128.9 on SHA, which outperforms all comparable methods.
>
> To further assess the model's domain generalizability, following those datasets suggested by you, we use the models trained on FSC-147 to directly evaluate their performance on IOCfish5k [B] and PASCAL VOC 2007 [C], which involve visually ambiguous and heavily occluded objects. Notice CLIP-Count (Jiang et al., ACM-MM2023), VLCounter (Kang et al., AAAI2024), CountGD (Amini-Naieni et al., NeurIPS2024), and T2ICount (Qian et al., CVPR2025) were originally fully-supervised trained in FSC-147 while ours is weakly-supervised trained.
> As shown in the table below, WS-COC achieves the best on these datasets, confirming its strong generalizability to handle visually ambiguous or occluded objects.
> In particular, WS-COC significantly outperforms other fully-supervised methods on the PASCAL VOC 2007 dataset, we suspect the reason is that similar data have already been incorporated into the pre-training corpus of LLaVA-OneVision. Consequently, WS-COC benefits from this prior knowledge and achieves outstanding performance.
>
> Moreover, we discuss the difference between [A,B,C] and our WS-COC: [A] proposes an Optimal Transport based strategy for crowd counting based on a CNN architecture, while [B] introduces a transformer-based architecture named IOCFormer for counting visually ambiguous fishes in underwater scenes. Both [A] and [B] are class-specific object counting methods. [C] focuses on counting common categories in daily-life images. It is a CNN-based class-agnostic object counting approach that mainly considers counting in sparse scenes (less than 10 objects). It should also be noted that [A,B,C] are fully-supervised methods. In contrast, our WS-COC leverages MLLMs to achieve weakly-supervised class-agnostic object counting, with a particular emphasis on dense and heavily occluded scenarios, for which we design specific optimization strategies.
>
> | Method | \| Supervision | \| IOCfish5K MAE  | IOCfish5K RMSE  | \| PASCAL VOC 2007 MAE  | PASCAL VOC 2007 RMSE  |
> |--------|----------------|--------------------|------------------|--------------------------|------------------------|
> | CLIP-Count (Jiang et al., ACM-MM2023) | \| point-level | \| 82.1  | 155.2 | \| 12.5  | 32.7 |
> | VLCounter (Kang et al., AAAI2024)     | \| point-level | \| 78.0  | 154.9 | \| 11.3  | 28.9 |
> | CountGD (Amini-Naieni et al., NeurIPS2024) | \| point-level | \| 67.4  | 144.8 | \| 8.7  | 21.7 |
> | T2ICount (Qian et al., CVPR2025)      | \| point-level | \| 63.3  | 136.1 | \| 9.9   | 23.5 |
> | **WS-COC**                            | \| image-level | \| **48.1** | **112.7** | \| **0.9** | **1.4** |

---

### Official Review · Reviewer_NtbD · 2025-11-08

**Soundness:** 2
**Presentation:** 3
**Contribution:** 2
**Rating:** 4
**Confidence:** 3

**Summary:**

The paper tackles the task of weakly supervised class-agnostic counting when only image-level counts are available. Proposed approach does not require point wise annotations. Instead of directly training an MLLM to emit a number, authors propose three strategies on top of Llava style MLLM: (1) Divide and Discern Dialogue Tuning (D3T): turn counting into a multi round binary search style judgement dialogue, then finalize the count once the range is narrow; (2) Compare and rank Count Optimization (CRCO): teach relative ordering by ranking 4 images sampled from different count bins per class; (3) Global and Local Counting Enhancement (GLCE): at inference, average the global prediction with a grid of local sub image predictions to balance global under counting and local over counting.

**Strengths:**

Proposed approach achieves reasonable performance without using any point wise annotations which are expensive to obtain.

**Weaknesses:**

1. Novelty of the proposed approach is somewhat limited. Each of the three proposed components rely on fairly standard heuristics. Novelty of the work is more in combining them for this setting than in any single algorithmic leap.
2. GLCE uses a 2×2 grid, with simple averaging of global and local. While GLCE is effective, it’s a heuristic and can be dataset sensitive.
3. Results are reported for only one MLLM backbone and model size. It's not clear if the approach will generalize to other MLLMs and model sizes.

**Questions:**

1. As an ablation study, I would be curious to see MLLM-Zero + GLCE and WS-COC-Base + GLCE perform. Since GLCE can be applied at test time, it is compatible with both MLLM-Zero and WS-COC-Base.
2. What are the main differences between the proposed approach and other contemporary works like https://arxiv.org/pdf/2412.00686v1 which also propose a divide and conquer approach for class agnostic counting, similar to GLCE.

---

> ### Author Response · Authors · 2025-11-22
> **Response to Weakness 1 of Reviewer NtbD**
>
> >  **Weakness 1:**
> > Novelty of the proposed approach is somewhat limited. Each of the three proposed components rely on fairly standard heuristics. Novelty of the work is more in combining them for this setting than in any single algorithmic leap.
>
> **Response:**
> We appreciate your comment. We would like to emphasize that the core novelty of our work lies in introducing the first MLLM-driven weakly-supervised framework for class-agnostic object counting. As being said, pre-trained MLLMs demonstrate limited counting ability, while the performance by directly fine-tuning them to count objects remains unsatisfactory. Hence, we introduce three simple yet effective strategies: divide-and-discern dialogue tuning, compare-and-rank count optimization, global-and-local counting enhancement. They are well motivated in object counting to progressively strengthen the MLLM's awareness in object counts. Extensive experiments over various benchmarks show that, by integrating every proposed strategy, we achieve substantial reduction in counting errors, attesting effectiveness, complementarity, and generalizability of the proposed strategies.
>
> Overall, we believe that, as long as our strategies are well motivated and empirically effective, they should not be disadvantaged by their simplicity/heuristics. Moreover, we would like to concur comments from other reviewers that:  1) Our approach of bootstrapping a generative MLLM for the counting task, rather than building upon a discriminative VLM, clearly distinguishes it from other existing works. As the first of its kind, the bootstrapping strategies have not been proposed before and are indeed novel for MLLM-driven object counting; 2) Our experiments, being rigorous and well structured, demonstrate that the proposed approach is even competitive with those of fully-supervised approaches.  All these make our work a novel and promising work in the object counting field.

---

> ### Author Response · Authors · 2025-11-22
> **Response to Weakness 2 of Reviewer NtbD**
>
> >  **Weakness 2:**
> > GLCE uses a 2×2 grid, with simple averaging of global and local. While GLCE is effective, it’s a heuristic and can be dataset sensitive.
>
> **Response:**
> Thanks.  GLCE was proposed based on a consistent observation across datasets: MLLMs tend to underestimate object counts in dense scenes. This is owing to the nature that MLLMs have mostly been pre-trained on images with sparse crowds and small number of objects.
>
> Hence, we argue that GLCE is empirically dataset insensitive. To alleviate your concern, we provide below the detailed ablation study of GLCE on various datasets, i.e. FSC-147, SHA, SHB, CARPK, and PUCPR+, where you can see its clear improvement on each of them. Furthermore, the choice of 2×2 grid concerns both efficiency and effectiveness in practice. As being said, due to the edge effect when aggregating sub-images, having smaller grids may potentially introduce more errors in the overall count. As illustrated in the table below, L=2 performs the best on FSC-147, SHB, and CARPK, while L=3 performs slightly better than L=2 on SHA and PUCPR+; not to mention that the computation cost has been doubled from L=2 to L=3.
>
>
> | Method                  | \| FSC-147 MAE  | FSC-147 RMSE  | \| SHA MAE  | SHA RMSE  | \| SHB MAE  | SHB RMSE  | \| CARPK MAE  | CARPK RMSE  | \| PUCPR+ MAE  | PUCPR+ RMSE  |
> |-------------------------|------------------|----------------|--------------|------------|--------------|------------|----------------|---------------|-----------------|----------------|
> | WS-COC w/o GLCE         | \| 15.72         | 105.25         | \| 147.1     | 263.9      | \| 41.7      | 64.3       | \| 13.78       | 20.99         | \| 52.76        | 67.32          |
> | WS-COC w/ GLCE (L=2)    | \| **13.91**     | **97.28**      | \| 128.9     | 232.9      | \| **34.2**  | **57.0**   | \| **10.39**   | 15.83         | \| 42.30        | 54.06          |
> | WS-COC w/ GLCE (L=3)    | \| 14.34         | 99.93          | \| **127.4** | **227.7**  | \| 38.5      | 57.7       | \| 11.14       | **15.64**     | \| **41.64**    | **53.95**      |
> | WS-COC w/ GLCE (L=4)    | \| 15.49         | 101.85         | \| 138.2     | 258.9      | \| 38.4      | 59.5       | \| 13.12       | 17.15         | \| 46.35        | 58.14          |

---

> ### Author Response · Authors · 2025-11-22
> **Response to Weakness 3 of Reviewer NtbD**
>
> >  **Weakness 3:**
> > Results are reported for only one MLLM backbone and model size. It's not clear if the approach will generalize to other MLLMs and model sizes.
>
> **Response:**
> Thanks.  In the table below, we report the performance of WS-COC over different MLLM backbones and model sizes on the FSC-147 dataset, including LLaVA-OneVision-7B (Li et al., 2024), LLaVA-OneVision-0.5B, Qwen3-VL-4B (Yang et al., 2025), Qwen3-VL-8B, DeepSeek-VL2-3B (Wu et al., 2024) and DeepSeek-VL2-16B. We can see that using larger models generally achieves better performance: compared with our default LLaVA-OneVision-7B, when using Qwen3-VL-8B with a similar model size, the performance is actually comparable; on the other hand, we can even achieve better performance when using a larger model (DeepSeek-VL2-16B). Overall, considering the efficiency and accuracy trade-off, we choose LLaVA-OneVision-7B by default. All these results show that WS-COC generalizes well across different MLLM backbones.
>
>
> | Method                          | \| VAL MAE  | VAL RMSE  | \| TEST MAE  | TEST RMSE  |
> |---------------------------------|--------------|------------|---------------|-------------|
> | WS-COC (DeepSeek-VL2-3B)        | \| 16.47     | 59.23      | \| 16.26      | 103.23      |
> | WS-COC (DeepSeek-VL2-16B)       | \| **14.45** | 56.71      | \| **13.67**  | **95.51**   |
> | WS-COC (Qwen3-VL-4B)            | \| 16.68     | 58.75      | \| 15.51      | 99.21       |
> | WS-COC (Qwen3-VL-8B)            | \| 15.18     | 55.90      | \| 14.13      | 97.23       |
> | WS-COC (LLaVA-OneVision-0.5B)   | \| 17.05     | 60.87      | \| 17.32      | 105.89      |
> | WS-COC (LLaVA-OneVision-7B)     | \| 14.77     | **54.24**  | \| 13.91      | 97.28       |

---

> ### Author Response · Authors · 2025-11-22
> **Response to Question 1 of Reviewer NtbD**
>
> >  **Question 1:**
> > As an ablation study, I would be curious to see MLLM-Zero + GLCE and WS-COC-Base + GLCE perform. Since GLCE can be applied at test time, it is compatible with both MLLM-Zero and WS-COC-Base.
>
> **Response:**
> As suggested, we report the results of applying GLCE to both MLLM-Zero and WS-COC-Base in the table below on the FSC-147 dataset.  The results show that GLCE consistently improves their performance.  These findings further validate the effectiveness of GLCE.
>
> | Method                  | \| VAL MAE  | VAL RMSE  | \| TEST MAE  | TEST RMSE  |
> |--------------------------|--------------|------------|---------------|-------------|
> | MLLM-Zero               | \| 38.92     | 119.26     | \| 38.19      | 145.42      |
> | MLLM-Zero w/ GLCE       | \| 33.49     | 102.83     | \| 32.65      | 128.70      |
> | WS-COC-Base             | \| 21.70     | 87.53      | \| 21.08      | 122.18      |
> | WS-COC-Base w/ GLCE     | \| 19.36     | 75.96      | \| 18.85      | 111.53      |
> | WS-COC                  | \| **14.77** | **54.24**  | \| **13.91**  | **97.28**   |

---

> ### Author Response · Authors · 2025-11-22
> **Response to Question 2 of Reviewer NtbD**
>
> >  **Question 2:**
> > What are the main differences between the proposed approach and other contemporary works like [https://arxiv.org/pdf/2412.00686v1](https://arxiv.org/pdf/2412.00686v1) which also propose a divide and conquer approach for class agnostic counting, similar to GLCE.
>
> **Response:**
> Thanks for your question. LVLM-COUNT (Qharabagh et al., 2024) relies on additional pretrained models (GroundingDINO and SAM) to generate region proposals and object masks to divide the image for counting. This results in a more complex pipeline and a strong dependency on the detection/segmentation quality (errors can directly propagate to the final counting result). In contrast, our GLCE does not require external modules. It simply partitions the image into grids and aggregates local predictions.
>
>
> In the table below we implement the LVLM-COUNT’s strategy within our framework (denoted as WS-COC w/ $\mathrm{GLCE\_LVLM}$) on the FSC-147 dataset. We can see this variant’s performance is better than WS-COC w/o GLCE but lower than our default WS-COC with GLCE, suggesting GLCE is simpler but more effective. Especially, since GLCE is specifically designed for improving object counting in dense scenes, we evaluate the models on images with more than 100 objects from FSC-147, i.e., VAL SET (dense) and TEST SET (dense). The performance difference between WS-COC and WS-COC w/ $\mathrm{GLCE\_LVLM}$ indeed gets bigger, e.g., WS-COC w/ $\mathrm{GLCE\_LVLM}$ increases the MAE relative to WS-COC by +10.2% on the entire VAL SET, while by +13.7% on the VAL SET (dense).
>
>
> | Method                 | \| VAL MAE  | VAL RMSE   | \|VAL (dense) MAE  | VAL (dense) RMSE  |\| TEST MAE  | TEST RMSE  |\| TEST (dense) MAE  | TEST (dense) RMSE  |
> |---------------------------|-----------|----------------|--------------------|-----------------------|-------------|----------------|---------------------|----------------------|
> | WS-COC w/o GLCE       |  \| 16.64     | 65.08      | \|  80.33              | 161.24              |\| 15.72       | 105.25       |\| 67.23               | 275.56               |
> | WS-COC w/ $\mathrm{GLCE\_LVLM}$ |  \|16.28    | 63.17       |  \|78.67              | 156.93          | \|15.19       | 104.30      |\| 64.50               | 269.63               |
> | WS-COC                | \|  **14.77** | **54.24**  |\|   **69.18**          | **143.22**     |\| **13.91**   | **97.28**    |\| **54.37**           | **238.89**           |

---

### Author Response · Authors · 2025-11-23
**Response to reviewers**

We thank the reviewers for their positive feedback and constructive comments to our work. We have very carefully considered all reviewers’ comments and have made every effort to address them in below. The corresponding revision will be incorporated into the revised manuscript accordingly.

---

### Author Response · Authors · 2025-12-03
**Revision summary**

We thank the area chairs for their efforts in handling our submission and thank the anonymous reviewers for their positive feedback and constructive comments. We propose the first MLLM-driven weakly-supervised framework for class-agnostic object counting (WS-COC). As the reviewers say, our approach of bootstrapping a generative MLLM for the counting task, rather than building upon a discriminative VLM with a specialized counting head, is a novel and promising direction [Reviewer GPEv and XCqq]. The three strategies we introduce are conceptually simple yet effective, with clear motivations [Reviewer 3dgp and QMGG]. The experimental design is rigorous and comprehensive [Reviewer GPEv, QMGG and XCqq], and our approach achieves performance competitive with fully-supervised approaches, marking a notable contribution to the field [Reviewer QMGG and XCqq]. We have very carefully considered all reviewers’ comments and have made every effort to address them. The corresponding revisions have now been integrated into the revised manuscript, highlighted in blue, and have been uploaded. Below we summarize the revisions:

**Backbone choice.** We add experimental results of our WS-COC based on other MLLMs, such as Qwen3-VL-4B, Qwen3-VL-8B, and DeepSeek-VL2-16B, in Tab. 6 of Sec. 5.3. All variants achieve very good results, confirming that WS-COC generalizes well over different MLLM backbones.

**Visualization.** We provide qualitative comparison between WS-COC and state-of-the-art methods in Fig. 3, demonstrating the impressive performance achieved by WS-COCO. In addition, we offer the visualization of attention maps for WS-COC and our baseline model (WS-COC-Base) in Fig. 5, which illustrates that our strategies progressively refine MLLMs to focus towards objects that are to be counted.

**Model complexity.** We add comparison of training cost, inference speed, and memory usage between state-of-the-art fully-supervised object counting methods and our WS-COC at the end of Sec. 5.2. The results clarify that although our approach leverages MLLMs, the adoption of parameter-efficient fine-tuning significantly reduces the computation cost. In fact, the main cost stems from the inherent design of MLLMs, while our proposed strategies introduce negligible additional cost. Overall, WS-COC remains competitive among other representative methods.

**Datasets.** We provide additional cross-dataset evaluation in Tab. D of Appendix A using IOCfish5K and VOC PASCAL 2007. The results demonstrate WS-COC’s strong generalizability across diverse domains.

**Analysis.** We first expand failure case analysis in Tab. E and Fig. D of Appendix B. We observe that all methods including ours suffer in extremely dense scenes while further analysis reveals that counting errors are increased approximately with an inverse proportion to the average object size per image. Next, we provide additional analysis on the dataset sensitivity and image partition scheme of the proposed global-and-local counting enhancement (GLCE) in Tab. A and Tab. C of Appendix A. Last, we conduct additional ablation study regarding the sampling schemes in compare-and-rank count optimization (CRCO) in Fig. B and Tab. B of Appendix A. All these results demonstrate the robustness of our proposed CRCO and GLCE.

---

### Meta-Review · Area_Chair_CPKf · 2026-01-12

**Summary:**

The only reviewer who rejected the paper has mostly subjective concerns. The proposed approach seems to offer an improvement, so I recommend accepting the paper.

**Reviewer Concerns:**

NtbD's concerns seem highly subjective.

**Reviewer Scores:**

I can't provide this on behalf of the reviewers.

---

### Decision · Program_Chairs · 2026-01-26

Accept (Poster)